# Cold-Start Personalization via Bayesian Adaptive Questioning

**Avinandan Bose** [*1 2]  **Shuyue Stella Li** [*1 2]  **Faeze Brahman** [3]  **Pang Wei Koh** [2 3]  **Simon Shaolei Du** [2]
**Yulia Tsvetkov** [§ 2]  **Maryam Fazel** [§ 2 4]  **Lin Xiao** [§ 1]  **Asli Celikyilmaz** [§ 1]

## Abstract

Cold-start personalization requires inferring preferences from minimal interaction when no user-specific historical data is available. The space of possible preferences is vast, yet users care about only a sparse subset and rarely articulate them upfront; combined with limited interaction budgets, this makes preference elicitation challenging. Our key insight is that preferences exhibit predictable structure across populations; e.g., users who want detailed explanations often also value worked examples. We propose CAPE (**C**old-start **A**daptive **P**reference **E**licitation), which learns a structured world model of preference correlations offline from complete profiles, then performs training-free Bayesian inference online to select informative questions and predict complete preference profiles, including dimensions never asked about. Even simple belief model instantiations (e.g., linear regression) substantially outperform end-to-end RL. Across medical, mathematical, social, and commonsense reasoning, CAPE achieves 80.8% alignment with ground-truth user preferences versus 68.5% for RL, requires 3-5× fewer interactions, and adapts twice as often. Our contribution is a principled decomposition of cold-start personalization that makes Bayesian preference elicitation practical at scale for LLM systems.

## 1. Introduction

Cold-start personalization—inferring user preferences from minimal interaction when no user-specific historical data is available—is fundamental to user satisfaction yet poorly addressed by current systems. Consider two users who ask

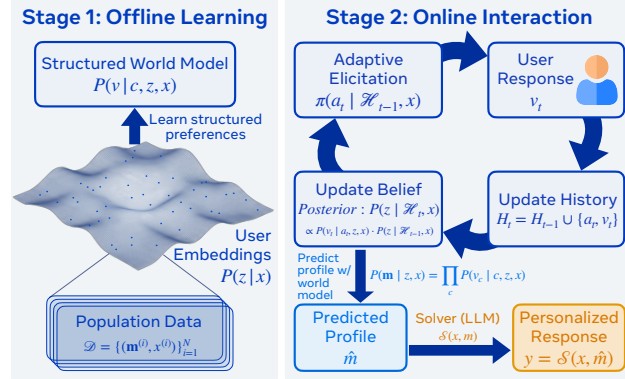

*Figure 1.* Overview of CAPE. **Offline:** We learn a structured world model from population data capturing preference correlations through latent user embeddings. **Online:** For new users, we adaptively select informative questions, update beliefs after each response, and predict the full preference profile, including preferences we never asked about, to generate personalized responses.

an AI assistant, "What is the best way to treat lower back pain?" A physical therapist wants a clinically precise answer with anatomical detail and evidence-based protocols; a patient with no medical background wants simple, actionable guidance in plain language. The optimal response for one actively hinders the other. The challenge for the assistant is that neither user will volunteer this upfront—the assistant must elicit it. Furthermore, cold-start is not only a first-session problem, since it recurs whenever a task's preference-relevant dimensions fall outside the span of prior interactions, even for users with extensive history. In a large-scale evaluation spanning medical, mathematical, and commonsense reasoning, Li et al. (2026) find that frontier models fail to ask appropriate clarifying questions even when explicitly prompted, with 29% of elicitation attempts worsening alignment versus generic responses.

The challenge is compounded by sparsity: while tasks admit dozens of preference dimensions (e.g., detail level, use of analogies, assumed background, emotional tone), individual users typically care about only a small subset (Li et al., 2025c; 2026). With limited interaction budgets (Radlinski et al., 2019), non-strategic questioning will likely miss the few dimensions each user values. This necessitates adaptive elicitation: selecting questions based on observed responses

---

[*]Equal contribution [§]Equal advising [1]Meta Superintelligence Labs [2]University of Washington [3]Allen Institute for AI [4]Amazon Inc.. Correspondence to: Avinandan Bose <avibose@cs.washington.edu>, Shuyue Stella Li <stelli@cs.washington.edu>.

*Proceedings of the 43rd International Conference on Machine Learning*, Seoul, South Korea. PMLR 306, 2026. Copyright 2026 by the author(s).

to efficiently narrow uncertainty about each user's preferences. This can be formalized as a partially observable Markov decision process (POMDP), where the user's preference profile is the hidden state, questions are actions, and responses are observations (Boutilier, 2002; Young et al., 2013). Furthermore, learning a questioning policy via reinforcement learning (RL) struggles because it must simultaneously discover preference structure and learn an adaptive policy from sparse multi-turn rewards (Wang et al., 2025; Wei et al., 2025; Xu et al., 2025).

Our key insight is that user preferences exhibit predictable structure across populations: users who want detailed explanations often also value worked examples and technical depth. These correlations constitute a *world model* of the preference environment. By learning this world model offline from complete preference profiles, we enable inference about unobserved preferences from partial observations, in which observing one dimension updates beliefs about all others through the learned correlation structure. This separation is critical: learning preference correlations from complete profiles requires only polynomial samples with dense per-criterion supervision, whereas RL requires sample complexity exponential in the interaction budget.

We propose CAPE (**P**reference **E**licitation with **P**riors), which decomposes cold-start elicitation into offline structure learning and online Bayesian inference, sidestepping credit assignment entirely (Figure 1). Offline, we learn a structured world model from complete preference profiles, capturing how preference dimensions correlate through latent user embeddings. Online, for each new user, we maintain a posterior over their latent embedding and update it via Bayes' rule as responses arrive, with questions selected to maximize information gain about the user's complete profile, including preferences never directly asked about. The framework is training-free at test time and modular, allowing any black-box LLM to serve as the downstream solver. It admits sample complexity guarantees polynomial in the number of criteria and independent of the interaction budget $T$, whereas RL scales exponentially (Appendix A).

We evaluate CAPE on four reasoning domains: medical diagnosis (MedQA), mathematics (AIME), commonsense reasoning (CSQA), and social reasoning (SocialIQA). These tasks are unambiguous in what to solve but admit diverse user preferences in how to reason and respond. CAPE achieves 80.8% alignment between generated responses and users' stated preferences versus 68.5% for RL (GRPO), with 3–5× fewer interactions. When two users give different answers to the same question, CAPE changes its follow-up question 39–62% of the time versus 0–28% for RL. Ablations confirm that the learned world model is the primary driver of these gains, with adaptive question selection saving one additional question per interaction. CAPE achieves this

with ≈10K parameters (Bayesian linear regression) versus 8B for RL baselines, demonstrating that the bottleneck in cold-start elicitation is not model capacity but whether the method exploits the factored structure of preference data.

Our contributions include:

1. We present a principled system decomposition for cold-start personalization into offline world model learning and online Bayesian inference, making Bayesian elicitation scalable.
2. We introduce CAPE, a modular elicitation framework that combines latent preference models with information-theoretic question selection, requiring no retraining at test time and integrating with black-box LLM solvers.
3. We provide an extensive empirical evaluation across four reasoning domains showing that this decomposition yields substantially higher preference alignment and interaction efficiency than end-to-end policy learning under sparse feedback.

## 2. Problem Formulation

We consider an interactive setting where an assistant must personalize its response to a user's task by inferring their task-specific preferences. The task $x$ may be unambiguous (e.g., "solve this equation"), but how to reason through and respond is not: different users prefer different tradeoffs across criteria such as detail, rigor, pedagogy, or brevity. These preferences are latent: users rarely state them upfront, and the relevant criteria vary across tasks.

**Preference profiles.** We represent a user's preferences as a profile $\mathbf{m}^* \subseteq \{(c, v) : c \in \mathcal{C}(x), v \in \mathcal{V}(c)\}$, recording which criteria the user cares about and what values they assign. The criteria set $\mathcal{C}(x)$ is a spanning set of relevant preference dimensions for a task. Each criterion $c$ admits a set of possible preference values $\mathcal{V}(c)$, which may represent categorical options, ordinal levels, or natural language descriptions. The set $\mathcal{V}(c)$ includes a distinguished element representing indifference. In practice, users care about only a small subset of available criteria, so $|\mathbf{m}^*| \ll |\mathcal{C}(x)|$.

**Elicitation as a POMDP.** The assistant interacts with the user to uncover their preferences before generating a response. We formalize this as a partially observable Markov decision process (POMDP):

- **State:** Hidden state is the user's preference profile $\mathbf{m}^*$.
- **Actions:** At each turn $t$, the assistant's action is to select a criterion to query: $a_t \in \mathcal{C}(x) \setminus \{a_1, \ldots, a_{t-1}\}$.
- **Observations:** The user responds with their preference $v_t \in \mathcal{V}(a_t)$ for the queried criterion, yielding observation $o_t = (a_t, v_t)$.
- **History:** The observable history after $t$ turns is $\mathcal{H}_t =$

$\{(a_1, v_1), \ldots, (a_t, v_t)\}$.
- **Horizon:** The assistant has a fixed budget of $T$ questions.
- **Policy:** A policy $\pi(a \mid \mathcal{H}_t)$ maps the current history to a distribution over which criterion to ask next.

This frames preference elicitation as exploring a structured environment where user preferences are the hidden state and questions and user responses provide partial observations.

**Profile prediction and response generation.** After $T$ turns, the assistant predicts a complete preference profile $\hat{\mathbf{m}}$ from the partial observations $\mathcal{H}_T$, including preferences for criteria that were never queried. The predicted profile then conditions the downstream response $y = \mathcal{S}(x, \hat{\mathbf{m}})$, where $\mathcal{S}$ is a fixed LLM that generates a response given the task and predicted preferences.

**Objective.** The quality of the final assistant response is measured by its preference alignment with the user's true preferences: $\text{PREFALIGN}(y, \mathbf{m}^*)$ (Li et al., 2026). This score is *user-conditioned*: the same response $y$ may score highly for one user and poorly for another. This rules out one-size-fits-all approaches because a response optimized for the population mean $\bar{\mathbf{m}} = \mathbb{E}[\mathbf{m}^*]$ will fail users whose preferences deviate from the average. Furthermore, preferences are *task-conditioned*: the same user may prefer rigorous derivations for mathematics but intuitive explanations for biology. The assistant faces a cold-start problem for each new task.

Our goal is to learn a policy that maximizes expected preference alignment across a distribution of users and tasks:

$$\max_{\pi} \; \mathbb{E}_{x \sim \mathcal{D}, \, \mathbf{m}^* \sim P(\mathbf{m}|x)} \, \mathbb{E}_{\tau \sim \pi} \left[ \text{PREFALIGN}(y, \mathbf{m}^*) \right],$$
(1)

where the trajectory $\tau = (\mathcal{H}_T, \hat{\mathbf{m}}, y)$ is induced by the policy's question selections, the resulting observations, the predicted profile, and the solver's response.

**Learning from world models versus sparse rewards.** An effective elicitation policy requires a world model: knowledge of how preference dimensions correlate across users. Reinforcement learning learns this world model implicitly while learning the questioning policy, using only sparse terminal rewards. The following result shows this joint learning problem has exponential sample complexity:

**Proposition 2.1** (Abbreviated). *Learning an effective questioning policy via RL requires exploring the space of question sequences, which grows combinatorially in $|\mathcal{C}(x)|$ (# of criteria). With only terminal feedback, sample complexity scales exponentially in the budget $T$. In contrast, learning preference correlations from complete profiles requires samples polynomial in $|\mathcal{C}(x)|$ and independent of $T$.*

Appendix A provides a formalized version of this argument

under standard assumptions. This exponential scaling arises because RL must solve credit assignment across sequential decisions without explicit access to correlation structures.

## 3. Methodology

We introduce CAPE, which is a framework that decomposes cold-start preference elicitation into two components: (i) a *belief model* that captures population-level preference structure from complete profiles (Figure 1 left), and (ii) a *selection strategy* that uses this structure to adaptively query users under an interaction budget (Figure 1 right). CAPE is training-free in that it performs no parameter updates for new users; adaptation occurs solely through Bayesian updates using a fixed, offline-learned belief model and can be plug-and-play with any solver mode for response generation.

**World model.** We model statistical regularities and correlations in user preferences through a task-conditioned population distribution $P(\mathbf{m} \mid x)$. By learning this distribution from complete profiles offline, we can construct a prior $P(v|c, x)$ for each criterion that encodes population-level correlations. At test time, partial observations allow inference over unobserved criteria via this learned structure.

### 3.1. Belief Models

The belief model maintains a posterior over user preferences given interaction history. Recall from Section 2 that $\mathcal{H}_t = \{(a_1, v_1), \ldots, (a_t, v_t)\}$ denotes the observed history after $t$ interactions, where $a_i \in \mathcal{C}(x)$ is the criterion queried at turn $i$ and $v_i \in \mathcal{V}(a_i)$ is the user's response. For each unqueried criterion $c \in \mathcal{C}(x) \setminus \{a_1, \ldots, a_t\}$, the belief model outputs

$$b_t(v \mid c) := P(v \mid c, x, \mathcal{H}_t), \quad v \in \mathcal{V}(c). \quad (2)$$

This conditional distribution captures both the predicted value and the uncertainty in that prediction.

**Latent variable formulation.** We assume a latent variable $z$ (user embedding) mediates dependencies between criteria, yielding

$$P(\mathbf{m} \mid z, x) = \prod_c P(v_c \mid c, z, x). \quad (3)$$

This structure provides three benefits: (i) compact representation of the joint preference distribution without modeling exponentially many configurations; (ii) a coherent belief state where the posterior $P(z \mid \mathcal{H}_t, x)$ summarizes everything about the user after $t$ observations; (iii) well-defined information gain—querying any criterion informs $z$, which propagates to all others. When the posterior is tractable, information gain is computed exactly; otherwise, it can be estimated via sampling or variational methods.

**Offline learning.** Given a dataset $\mathcal{D} = \{(\mathbf{m}^{(i)}, x^{(i)})\}_{i=1}^{N}$ of complete preference profiles, we learn task-conditioned priors

$$b_0(v \mid c) := P(v \mid c, x) \quad \text{for } v \in \mathcal{V}(c). \quad (4)$$

These encode population-level preference correlations before any user-specific interaction.

**Bayesian updates.** After observing $(a_t, v_t)$, the posterior over the latent variable $z$ updates via Bayes' rule:

$$P(z \mid \mathcal{H}_t, x) \propto P(v_t \mid a_t, z, x) \cdot P(z \mid \mathcal{H}_{t-1}, x), \quad (5)$$

which induces updated predictive distributions $b_t(v \mid c)$ for all unobserved criteria. This Bayesian updating allows the model to maintain beliefs over individual users while leveraging the world model learned from population data.

**Instantiations.** The framework admits any belief model that maintains a posterior over $z$. The input representation $\phi(\mathcal{H}_t)$ can range from simple indicator encodings to neural networks, while the output head maintains a structured form (e.g., Gaussian, mixture) for tractable information gain. This separation allows expressive representations while preserving principled uncertainty quantification. We use two collaborative-filtering instantiations.

*Example 1: Bayesian Linear Regression.* The embedding $z = \phi(\mathcal{H}_t)$ encodes observed criteria and values. Each criterion is modeled as

$$v_c = \boldsymbol{\beta}_c^\top z + \epsilon, \quad \epsilon \sim \mathcal{N}(0, \sigma^2), \quad (6)$$

with the weights $\boldsymbol{\beta}_c$ learned offline. Predictions integrate over the learned posterior: $b_t(v \mid c) = \int P(v \mid \boldsymbol{\beta}_c, z) P(\boldsymbol{\beta}_c \mid \mathcal{D}) \, d\boldsymbol{\beta}_c$.

*Example 2: Gaussian Mixture Model.* The latent variable $z \in \{1, \ldots, K\}$ indexes user types with task-conditioned priors $P(z \mid x)$ and type-specific likelihoods $P(v \mid c, z, x)$. The posterior is

$$P(z \mid \mathcal{H}_t, x) \propto P(z \mid x) \prod_{(a_i, v_i) \in \mathcal{H}_t} P(v_i \mid a_i, z, x). \quad (7)$$

Predictions marginalize over user types: $b_t(v \mid c) = \sum_z P(v \mid c, z, x) P(z \mid \mathcal{H}_t, x)$.

Both models admit closed-form posteriors; Appendix B.1 presents derivations and discusses methods for approximating posteriors for models that do not have tractable forms.

### 3.2. Adaptive Question Selection

The selection strategy chooses which criterion to query next given the current posterior. At turn $t$, the system must select which criterion to query next: $a_t \in \mathcal{C}(x) \setminus \{a_1, \ldots, a_{t-1}\}$.

**Strategies.** Any function that scores candidate criteria given the current posterior $P(z \mid \mathcal{H}_{t-1}, x)$ can be used. Strategies differ in whether they are are adaptive (i.e., conditioning on user responses) through this posterior. We consider three examples.

*Example 1: Random Selection.* A criterion is chosen uniformly among those remaining:

$$a_t \sim \text{Uniform}(\mathcal{C}(x) \setminus \{a_1, \ldots, a_{t-1}\}). \quad (8)$$

This is a non-adaptive strategy that ignores $H_{t-1}$, though the belief model still updates for the final prediction of $\hat{m}$. This isolates the contribution of adaptive selection from belief model inference.

*Example 2: Uncertainty Sampling.* The criterion with highest marginal entropy ($\mathbb{H}[\cdot]$) is queried:

$$a_t = \arg \max_{c \in \mathcal{C}(x) \setminus \{a_1, \ldots, a_{t-1}\}} \mathbb{H}[b_t(\cdot \mid c)]. \quad (9)$$

Since $b_t(v \mid c) = P(v \mid c, x, H_t)$ depends on the observed history through the posterior over $z$, this strategy is adaptive to user responses. This reduces uncertainty where it is highest but ignores correlations.

*Example 3: Information Gain.* The criterion that maximally reduces uncertainty about the latent variable is queried:

$$a_t = \arg \max_{c \in \mathcal{C}(x) \setminus \{a_1, \ldots, a_{t-1}\}} I(v_c; z \mid \mathcal{H}_t, x), \text{ where} \quad (10)$$

$$I(v_c; z \mid \mathcal{H}_t, x) = \mathbb{H}[z \mid \mathcal{H}_t, x] - \mathbb{E}_{v \sim b_t(\cdot \mid c)} \mathbb{H}[z \mid \mathcal{H}_t, (c, v), x] \quad (11)$$

is the mutual information between the response to criterion $c$ and the latent $z$. Rather than scoring criteria by individual uncertainty, this adaptive strategy scores by informativeness about the latent structure connecting all preferences. Stochastic variants of $\arg \max$ are also possible (Bose et al., 2024a).

Entropy and information gain are computed in closed form with full derivations in Appendix B.2, along with bounds and approximation methods for when posteriors are not tractable. The $\arg \max$ is computed by enumeration over remaining criteria, which is tractable since $|\mathcal{C}(x)|$ is typically small (10-20). The quantities for entropy and information-gain can be computed independently for each criteria and thus can be computed in parallel.

**Elicitation procedure.** Algorithm 1 summarizes CAPE. The policy $\pi(a \mid \mathcal{H}_t, x)$ is implicitly defined by combining the belief update with the selection strategy.

**Solver.** The solver $\mathcal{S}$ is treated as a fixed black box that generates a response given task $x$ and predicted preference profile $\hat{m}$. This modular design separates elicitation from response generation, allowing CAPE to focus on efficient preference inference.

**Algorithm 1** CAPE

**Require:** Task $x$, budget $T$, belief model conditional on user latent $z$, strategy SELECT, solver $\mathcal{S}$
**Ensure:** Personalized response $y$
1: $\mathcal{C} \leftarrow \mathcal{C}(x)$
2: $P(z|x) \leftarrow$ INITIALIZEPRIOR$(x)$
3: $\mathcal{H}_0 \leftarrow \emptyset$
4: **for** $t = 1, \ldots, T$ **do**
5:    $a_t \leftarrow$ SELECT$(P(z \mid \mathcal{H}_{t-1}), \mathcal{C} \setminus \{a_1, \ldots, a_{t-1}\})$
6:    $v_t \leftarrow$ QUERYUSER$(a_t)$
7:    $\mathcal{H}_t \leftarrow \mathcal{H}_{t-1} \cup \{(a_t, v_t)\}$
8:    $P(z \mid \mathcal{H}_t, x) \leftarrow$ UPDATEPOSTERIOR$(P(z \mid \mathcal{H}_{t-1}, x) \times P(v_t \mid a_t, z, x))$
9: **end for**
10: $\hat{\mathbf{m}} \leftarrow$ PREDICTPROFILE$(P(z \mid \mathcal{H}_T))$
11: $y \leftarrow \mathcal{S}(x, \hat{\mathbf{m}})$
12: RETURN   $y$

## 4. Experiments

### 4.1. Experimental Setup

We design experiments to test our central hypothesis: *effective preference elicitation requires learning population-level preference structure separately from adaptive individual inference.* Our experiments address three questions: (1) Does CAPE achieve better preference alignment? (2) Why does it outperform? (3) Which components are essential?

**Datasets.** We evaluate on four benchmarks covering diverse reasoning domains from PrefDisco (Li et al., 2026): MedQA (medical reasoning), AIME (mathematical problem-solving), CommonsenseQA (commonsense reasoning), and SocialIQA (social reasoning). Each domain contains 100 problems, with 50 users per problem. We filter the benchmark to ensure a challenging, long-tail distribution of preferences. We remove common cross-sample criteria (present in $\geq 10\%$ of tasks) to focus strictly on task-specific cold-start discovery, and exclude extremely rare criteria ($\leq 3$ users) to ensure statistical validity. This results in 20–30 diverse criteria per task. On average, users care about only 2–4 criteria. This sparsity creates a sparse discovery problem: since $\mathcal{C}(x)$ is a superset of user needs, random querying is ineffective, requiring our model to leverage learned correlations to prune the search space. We use an 80/20 train/test split at the problem level, so evaluation is performed on tasks never seen during training.

All evaluation settings and protocols, including criteria realism, user simulation and judge reliability, strictly follow the PrefDisco benchmark (Li et al., 2026), which provides extensive human evaluation confirming reliability. We adopts these protocols to ensure comparability and reproducibility.

**User simulation.** We evaluate under a passive user protocol: users answer minimally without volunteering information. This reflects documented human-AI interaction patterns where users provide minimal information unless prompted (Sundar, 2020) and isolates the method's information-seeking capability rather than relying on user initiative. We explore other user types in Appendix C.1 and find higher risks of reward hacking with collaborative users.

**Protocol and information access.** All methods follow the same protocol: sequentially select $T = 5$ criteria to ask about, observe user responses, and output a predicted preference profile over all criteria. This predicted profile is then passed to a shared solver (GPT-4.1) to generate a personalized response; the solver is fixed across all methods, isolating the effect of preference elicitation (see Appendix C.3 for solver prompt). All methods select from the same set of available criteria and observe the same user responses during elicitation.

**Baselines.** We compare against state-of-the-art methods for interactive preference elicitation spanning the spectrum of supervision and adaptivity:

1. **Prompting**: Llama-3.1-8B-Instruct given criteria descriptions and preference level descriptions instructed to ask informative questions. Uses the same base model and system prompt as GRPO for fair comparison.

2. **CollabLLM**: Llama-3.1-8B-Instruct trained with offline-DPO using multiturn reward (Wu et al., 2025b), targeting general interactivity rather than preference elicitation.

3. **Population Average**: Uses population-mean preferences without elicitation, no interaction needed.

4. **GRPO**: Llama-3.1-8B-Instruct given criteria descriptions and preference level descriptions trained with GRPO (Shao et al., 2024) using terminal PrefAlign rewards on personalized response generated using solver. Best checkpoint selected on validation set.

5. **CAPE**: Bayesian Linear Regression belief model with adaptive acquisition (Information Gain or Uncertainty variants, selected per dataset on validation). Evaluated over 20 trials.

These methods differ primarily in supervision structure. **Population Average**, **CAPE**, and **GRPO** all observe complete user preference profiles, receiving equivalent supervision quantity but different in organization. Population Average and CAPE receives these as factored per-criterion labels, enabling direct estimation of preference structures; while GRPO observes them through interaction trajectories where user responses reveal the same underlying preferences, learning from which requires jointly discovering

preference structure and questioning policy from terminal rewards. **CollabLLM** trains on trajectories without preference labels; **Prompting** receives no task-specific supervision. LLM-based baselines (Prompting, CollabLLM, GRPO) additionally have access to natural language descriptions of each criterion during elicitation. Our experiments test a focused hypothesis: whether separating preference structure learning from online policy learning yields practical benefits over end-to-end approaches. Full implementation details appear in Appendix D.

**Evaluation metrics.**

1. *Preference alignment*: We use PrefAlign (Li et al., 2026), a weighted alignment score computed by rubric-based LLM judge (GPT-4.1) to assess how well the response matches the user's ground-truth preferences. We normalize scores relative to Generic (no preference information, score $S_{\text{generic}}$) and Oracle (complete ground-truth profile, score $S_{\text{oracle}}$), reporting % of Oracle:

$$\% \text{ of Oracle} = \frac{S_{\text{method}} - S_{\text{generic}}}{S_{\text{oracle}} - S_{\text{generic}}} \times 100. \quad (12)$$

2. *Query efficiency*: We measure how many turns a method requires to reach a given preference alignment threshold, capturing the user interaction cost.

3. *Adaptivity*: Given identical history $\mathcal{H}_{t-1}$ and current question $a_t$, when two users provide different answers $v_t$, how often does $a_{t+1}$ differ?

$$\text{Adaptivity} = P(a_{t+1}^{(1)} \neq a_{t+1}^{(2)} \mid \mathcal{H}_{t-1}, a_t, v_t^{(1)} \neq v_t^{(2)}). \quad (13)$$

Effective elicitation should adapt questions based on user responses; methods that ask fixed question sequences regardless of answers cannot personalize to individuals.

**4.2. CAPE Achieves Better Preference Alignment**

Table 1 presents preference alignment across all methods and datasets. CAPE consistently and significantly outperforms all baselines, achieving 77-87% of oracle performance compared to 55-76% for GRPO, 72-82% for population average, 20-26% for CollabLLM, and 18-31% for prompting. These results validate our hypothesis that separating structure learning from adaptive inference is effective. Additionally, CAPE achieves these gains with approximately 10K parameters (400 criteria per domain × 25 regression weights per criterion; see Appendix D for details), demonstrating that the bottleneck in preference elicitation is inference structure, not model capacity.

**4.3. Fine-Grained Performance Gain Analysis**

We decompose CAPE's gains into three factors: (1) inference from partial observations, (2) adaptive question selection, and (3) learned preference structure.

*Table 1.* Preference alignment (% of Oracle, higher is better). Generic = 0%, Oracle = 100%. Mean ± std over 20 trials.

| Method | MedQA | AIME | SocialIQA | CSQA |
|---|---|---|---|---|
| Prompting | 22.3±2.1 | 29.1±2.3 | 31.4±2.2 | 18.2±1.9 |
| CollabLLM | 20.3±1.2 | 26.4±1.3 | 24.8±1.0 | 20.0±2.0 |
| Pop. Average | 73.2±1.4 | 74.3±1.8 | 82.1±1.2 | 72.4±1.3 |
| GRPO | 71.4±2.8 | 76.2±2.4 | 71.3±3.1 | 55.2±2.9 |
| **CAPE** | **77.4**±1.2 | **80.1**±1.4 | **87.3**±1.8 | **78.2**±1.1 |

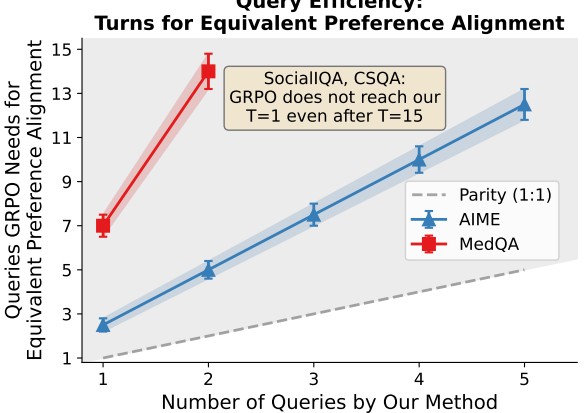

*Figure 2.* Query efficiency comparison between CAPE and GRPO. The x-axis shows the number of queries by CAPE; the y-axis shows how many queries GRPO needs to achieve the same preference alignment. Points above the dashed parity line indicate CAPE requires fewer queries than GRPO to achieve equivalent preference alignment. CAPE demonstrates 2.5× efficiency on AIME, 7× on MedQA, and >15× on SocialIQA and CSQA (where GRPO fails to match CAPE's T=1 performance even after 15 turns).

**4.3.1. QUERY EFFICIENCY**

To analyze this query efficiency of CAPE, we compare how many questions each method requires to reach equivalent preference alignment (Figure 2). We focus on comparison against GRPO as the strongest RL baseline for clarity.

CAPE achieves 2.5× efficiency on AIME, 7× on MedQA, and >15× on SocialIQA and CSQA, where GRPO fails to match CAPE's single-question performance even after 15 turns. By learning preference correlations offline from dense supervision, CAPE infers unobserved preferences from partial observations. GRPO must simultaneously learn correlations and questioning policy from sparse terminal rewards, requiring direct elicitation of each preference it uses.

**4.3.2. ADAPTIVE QUESTION SELECTION**

An effective elicitor should adapt to individual users rather than following a fixed sequence. We measure adaptivity: how often different user responses lead to different next questions (Eq. 13). Population Average is excluded as it asks no questions.

*Table 2.* Adaptivity: percent of cases where different user responses lead to different follow-up questions (higher is better) over 20 trials. CAPE adapts 2× more often than baselines.

| Dataset | Prompting | CollabLLM | GRPO | CAPE |
|---------|-----------|-----------|------|------|
| MedQA | 17.3±2.1 | 28.3±4.2 | 21.4±3.2 | **49.2**±3.1 |
| AIME | 29.1±2.4 | 44.3±3.1 | 0.0±0.0 | **39.4**±4.2 |
| SocialIQA | 28.7±3.1 | 75.0±12.5 | 27.8±2.9 | **61.8**±3.7 |
| CSQA | 22.4±2.3 | 37.0±4.8 | 21.6±2.8 | **43.3**±3.4 |

Table 2 shows CAPE adapts 2× more than baselines (39-62% vs 0-44%). GRPO tends exploit majority patterns and converge to static generic questioning. A fixed sequence of questions were generated to all users, resulting in an adaptivity score of 0% on AIME. GRPO's adaptivity correlates with cross-sample criteria overlap: on MedQA and AIME (5-8% overlap), it achieves 21-28% adaptivity and 71-76% alignment. On SocialIQA and CSQA (3% overlap), adaptivity remains low (22-28%) but performance drops to 55-71%, falling below population average. High overlap enables learning fixed question sequences for common patterns; low overlap yields sparse, diverse preferences where memorized sequences fail.

CollabLLM achieves 28-44% adaptivity by generating free-form follow-up questions. However, it targets general interactivity rather than preference elicitation, cycling through generic topics ("What is your background?" then "How do you prefer to learn?") rather than querying specific criteria informed by prior responses. This yields high adaptivity but only 20-26% preference alignment: changing questions is not sufficient; an effective elicitor must change questions *in a way that leverages learned preference structure* to target criteria the user actually cares about.

CAPE maintains 39-62% adaptivity across domains by learning correlations through latent variables rather than memorizing task-specific patterns. Figure 3 shows high adaptivity generally lead to better personalization (i.e., high preference alignment scores), with GRPO-AIME as an expected outlier with low adaptivity but high alignment given the high cross-sample preference overlap.

#### 4.3.3. QUALITATIVE COMPARISON

Table 3 compares GRPO and CAPE on two users with opposite preferences. GRPO asks identical questions regardless of responses, discovering 1/3 ground-truth preferences for each user. CAPE adapts: when User B rejects conversational tone, the world model infers formal preferences and pivots accordingly, discovering 3/3 preferences for both users.

This fixed-sequence behavior reflects a systematic failure mode. Table 10 (Appendix F) shows trajectories for a medical reasoning task across 8 users with entirely different ground-truth preferences. GRPO asks all 8 users the same

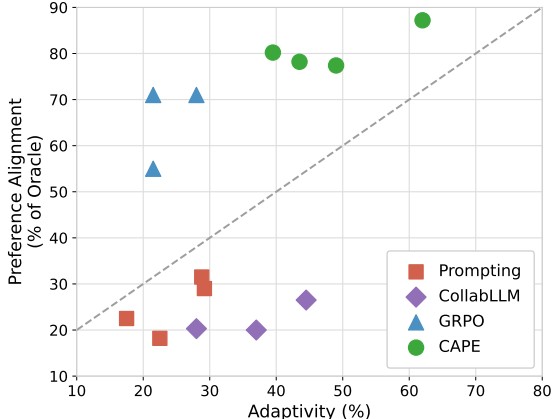

*Figure 3.* Adaptivity versus preference alignment across methods and datasets. Higher adaptivity is associated with better alignment.

first three questions (Hands-on Verification, Storytelling, Error Prevention), never adapting to observed responses. Critically, none of these questions correspond to any user's actual preferences: 4 out of 5 questions in each trajectory receive "I don't have a strong preference" responses, yet the policy does not update its strategy. This is the signature of policy collapse under sparse terminal reward: without per-criterion supervision, GRPO cannot determine which questions were informative, so it converges to a fixed sequence that performs adequately on average but fails any individual user.

CAPE avoids this failure by construction. Because the world model is trained with dense, per-criterion supervision offline, each observed response updates the posterior over the user's latent embedding and immediately shifts which criterion is most informative to query next. The adaptivity scores in Table 2 (39–62% for CAPE versus 0–28% for GRPO) are the aggregate signature of this mechanism; the trajectories in Tables 3 and 10 show what it looks like at the level of individual interactions.

#### 4.4. CAPE Components Ablations

We ablate CAPE's key components to validate our two-stage design: (1) offline world model learning and (2) online adaptive inference (Figure 4). In the first **offline world model learning** stage, removing the latent structure that captures correlations between preferences causes performance to plateau near population average, regardless of how many questions are asked (red). With learned correlations, each observation propagates information through the world model (blue, gray), confirming that population-level structure is essential for cold-start personalization. In the second **online adaptive inference** stage, Adaptive Querying (blue) consistently outperforms Random Selection (gray), closing 24% of the gap at $T=5$ compared to 21% for random. Adap-

*Table 3.* **Qualitative comparison on CommonsenseQA.** The task asks: "Why might someone walk to work instead of driving?" Two users with opposite style preferences answer elicitation questions. **User A** prefers casual, engaging explanations; their ground-truth preferences are Conversational Tone, Humor/Wit, Real-World Examples . **User B** prefers formal, thorough explanations; their ground-truth preferences are Formal Definitions, Structured Format, Comprehensive Coverage . Both methods begin with Q1 (Conversational Tone). User A responds "5" (strongly wants); User B responds "1" (strongly does not want). GRPO ignores this signal and asks both users the same fixed sequence. CAPE leverages the *negative* signal from User B: a "1" for Conversational Tone suggests this user likely prefers formality. The belief model updates accordingly, pivoting Q2 to Formal Definitions for User B (negatively correlated with casual tone) while asking User A about Humor (positively correlated). Shaded cells indicate ground-truth criteria. Scale: 1=strongly avoid, 5=strongly prefer, "no pref"=neutral.

| | | GRPO (Fixed Sequence) | | | CAPE (Adaptive) | | | |
| Turn | Criterion | User A (casual) | User B (formal) | User A Criterion | Resp. | User B Criterion | Resp. |
|---|---|---|---|---|---|---|---|
| Q1 | Conversational Tone | 5 | 1 | Conversational Tone | 5 | Conversational Tone | 1 |
| Q2 | Academic Citations | no pref | no pref | Humor/Wit | 4 | Formal Definitions | 5 |
| Q3 | Technical Terminology | 1 | 3 | Real-World Examples | 5 | Structured Format | 4 |
| Q4 | Abstract Principles | no pref | 2 | Storytelling | 3 | Comprehensive Coverage | 5 |
| Q5 | Theoretical Framework | no pref | no pref | Analogies | no pref | Precise Language | 3 |
| **Preferences Discovered** | | **1/3** | **0/3** | **3/3** | | **3/3** | |

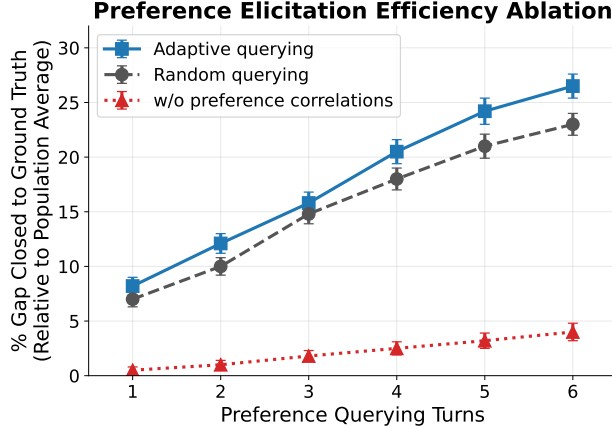

**Preference Elicitation Efficiency Ablation**

*Figure 4.* Ablation study showing (i) the importance of modeling preference correlations: without the latent structure that captures correlations between preferences (red), elicitation yields very slow improvement over population average as the number of interaction turns increases, compared to modeling preference correlations (blue, gray); and (ii) the benefit of adaptive querying: adaptive selection (blue) closes 24% of the gap at $T{=}5$, outperforming random querying (gray) at $T{=}6$, effectively saving one question.

tive querying at $T{=}5$ matches Random at $T{=}6$, saving one question per user interactions.

## 5. Related Work

**Cold-start inference and collaborative filtering.** The cold-start problem has been extensively studied in recommender systems, where collaborative filtering learns latent structure from population data to infer preferences for new users (Koren et al., 2009; Salakhutdinov & Mnih, 2008; Liang et al., 2018; Elahi et al., 2018), and active extensions select which items to query to reduce uncertainty (Boutilier et al., 2012; Jin & Si, 2012; Golbandi et al., 2011). Recent

work combines collaborative filtering with LLMs for recommendation (Zheng et al., 2024; Hou et al., 2024; Liao et al., 2023; Lin et al., 2024). Our setting differs in two key respects: preference dimensions vary per task rather than forming a fixed item catalog, and the output is a free-form natural language response whose quality depends on accurately capturing the user's full preference profile. Our belief models adapt collaborative filtering to this setting through adaptive Bayesian questioning, with Bayesian experimental design (Lindley, 1956; Chaloner & Verdinelli, 1995; Bickford Smith et al., 2023; Foster et al., 2021; Astudillo et al., 2023; Huber et al., 2025; Bose et al., 2024b) providing the information-theoretic foundation for question selection, and active learning (Settles, 2009; Lewis & Catlett, 1994; Seung et al., 1992) motivating the broader sample-efficient querying strategy. We cast elicitation as a POMDP (Kaelbling et al., 1998; Åström, 1965), with belief-state planning applied successfully to dialogue management (Young et al., 2013; Williams & Young, 2007; Thomson & Young, 2010) and online methods (Ross et al., 2008; Silver & Veness, 2010) planning through forward search in belief space. We separate world model learning (offline, dense supervision) from policy execution (online, Bayesian inference over individuals)—in contrast to model-based RL (Ha & Schmidhuber, 2018; Hafner et al., 2023; Schrittwieser et al., 2020), which learns world models and policies jointly from sparse rewards.

**Preference elicitation and personalization.** RLHF and DPO (Ouyang et al., 2022; Rafailov et al., 2023) align models to aggregated preferences without interactive discovery, and recent work adapts post-hoc through per-user reward modeling (Poddar et al., 2024; Li et al., 2025c; Bose et al., 2025). LLM-based preference elicitation has been explored in conversational recommenders (Austin et al., 2024; Montazeralghaem et al., 2025; He et al., 2023), combinatorial

auctions (Huang et al., 2025), and task specification (Li et al., 2023; Handa et al., 2024). SELM (Zhang et al., 2024) addresses active exploration in online RLHF but focuses on response diversity for reward modeling rather than structured preference discovery. PrefDisco (Li et al., 2026) demonstrates systematic failure of frontier models at proactive elicitation—29% of attempts worsen alignment versus generic responses, and recent benchmarks further document this gap (Afzoon et al., 2024; Zhao et al., 2025; Jiang et al., 2025). Our work addresses this through structured world model learning with dense, per-criterion supervision.

**Interactive and clarification-seeking systems.** A related line of work studies clarifying *task* ambiguity—resolving missing or underspecified information needed to complete a task. MediQ (Li et al., 2024) benchmarks question-asking in clinical settings where models must identify missing patient information; ALFA (Li et al., 2025b) trains models to ask higher-quality clarifying questions in clinical reasoning; QuestBench (Li et al., 2025a) and Radlinski et al. (2019) study whether models can identify the right question to resolve task ambiguity across reasoning and recommendation settings. CollabLLM (Wu et al., 2025a) and Pang et al. (2025) target general interactivity and chain-of-thought refinement respectively. Our setting is orthogonal: the task is unambiguous, but how to reason and respond depends on latent user preferences that vary across individuals. Eliciting these preferences requires discovering user objectives rather than filling task-information gaps, and the optimal response can differ fundamentally across users even for identical tasks.

## 6. Conclusion

We formalize the problem of cold-start preference elicitation and tackle it by decomposing the problem into offline learning of preference structure and online Bayesian inference. Our proposed method, CAPE, learns a structured world model of preference correlations from complete profiles with dense, per-criterion supervision, then performs training-free inference at test time to select informative questions and predict complete preference profiles from minimal interaction. By exploiting the factored structure of preference data that end-to-end RL discards, CAPE achieves substantially higher alignment with fewer interactions using simple belief models. The framework is training-free and modular, allowing the learned world model to augment any solver, including API-based or domain-specific models. Future work should extend to natural language elicitation, where questions and responses are free-form rather than structured over predefined criteria, and to automatic discovery of preference dimensions from data.

## Limitations

While our framework demonstrates significant gains in cold-start elicitation, several avenues for future research remain. We do not address longitudinal learning across sessions or helping users form preferences they have not yet considered.

We evaluate with simulated users following the validated protocols of PrefDisco (Li et al., 2026), which were designed to reflect empirically observed human–AI interaction patterns and have been validated against human judgments. Our evaluation therefore isolates the elicitation strategy itself rather than confounding effects from user verbosity or cooperation. Nonetheless, real-world deployment would introduce additional challenges such as noisy, inconsistent, or evolving preferences.

As the world model is trained on population-level preference data, it is crucial to ensure that the learned correlations do not encode or propagate social biases. Furthermore, while the inference happens locally at test-time to preserve user privacy, there might still exist privacy and security risks from the limited turns of interaction with the model. Real world applications must be augmented with rigorous de-identification and consent protocols.

## Impact Statement

This paper presents work whose goal is to advance personalized AI systems. Effective preference elicitation enables AI assistants to adapt to individual needs, improving user satisfaction and task outcomes across diverse populations.

Potential positive impacts include more equitable access to personalized assistance, as the method requires minimal user effort, no interaction history, and no technical expertise. The training-free design enables deployment without collecting per-user training data, enhancing privacy.

Potential risks include over-reliance on inferred preferences when users' actual needs differ, and bias amplification from population training data. Systems should provide mechanisms for users to correct inferred preferences and transparently disclose when responses are personalized based on inference rather than explicit statements.

## Reproducibility Statement

All experiments in this work use the open-source PrefDisco dataset. The full set of user simulation prompts is provided in Appendix C, and all hyperparameters and training configurations are detailed in Appendix D. We publicly release all code and evaluation scripts. [1]

---

[1] https://github.com/Avinandan22/PEP

## Acknowledgment

This research was developed in part with funding from the Defense Advanced Research Projects Agency's (DARPA) SciFy program (Agreement No. HR00112520300). The views expressed are those of the author and do not reflect the official policy or position of the Department of Defense or the U.S. Government. This material is based in part upon work supported by the Defense Advanced Research Projects Agency and the Air Force Research Laboratory, contract number(s): FA8650-23-C-7316. Any opinions, findings and conclusions, or recommendations expressed in this material are those of the author(s) and do not necessarily reflect the views of AFRL or DARPA. This research was supported by Coefficient Giving, the University of Washington Population Health Initiative, Amazon Health, the UW+Amazon Science Hub. SL gratefully acknowledges support from the Meta AIM program. The work of MF was supported in part by awards NSF CCF 2212261, NSF CCF 2312775, NSF TRIPODS II DMS-2023166, the Meta AIM program, and the Moorthy Family Professorship at UW.

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

# A. Query Complexity Analysis

We analyze the query complexity, the total number of user interactions required, for reinforcement learning versus belief model estimation. The key distinction is not the number of users, but the *structure of feedback* available from each interaction.

## A.1. Setup

Let $C = |\mathcal{C}(x)|$ denote the number of criteria, $T$ the elicitation budget, $K$ the number of latent user types, and $V = \max_c |\mathcal{V}(c)|$ the maximum number of preference values per criterion. We assume access to a simulator that can sample users from the population and respond to preference queries.

## A.2. Reinforcement Learning: Sparse Feedback

In RL, the agent learns by running episodes. Each episode consists of:

1. Sample a user with profile $\mathbf{m}^* \sim P(\mathbf{m} \mid x)$

2. Ask $T$ questions, receiving responses $v_1, \ldots, v_T$

3. Receive terminal reward $\text{PREFALIGN}(y, \mathbf{m}^*)$

**Queries per episode.** Each episode requires $T$ queries to the user.

**Feedback structure.** The agent receives $T$ preference values during elicitation, but these are *inputs* to the policy, not learning signals. The only supervision is the terminal reward, a single scalar summarizing how well the final response matched the user's preferences. This reward does not decompose: it provides no indication of which questions were informative or which were wasteful.

**The credit assignment problem.** To learn which questioning strategies work, RL must solve a credit assignment problem: given a terminal reward, determine which of the $T$ sequential decisions contributed positively (Sutton et al., 1998). With sparse, terminal-only feedback, this requires extensive exploration.

**Exploration complexity.** The number of distinct $T$-step questioning sequences is:

$$\frac{C!}{(C-T)!} = C \cdot (C-1) \cdot \ldots \cdot (C-T+1) = O(C^T) \tag{14}$$

For $C = 20$ and $T = 5$, this exceeds $10^6$ sequences. To determine which sequences are effective for which user types, RL must explore a substantial fraction of this space.

**Sample complexity lower bounds.** For episodic RL with horizon $T$ and sparse rewards, standard lower bounds show that the number of episodes required scales polynomially in the state-action space and the horizon (Azar et al., 2017; Jin et al., 2018; Domingues et al., 2021). In our setting, the effective state space (possible histories) grows as $O((C \cdot V)^T)$. Even with function approximation, policy gradient methods suffer from variance that scales with the horizon when rewards are sparse (Schulman et al., 2015; Papini et al., 2018), requiring:

$$N_{\text{episodes}} = \Omega\left(\frac{T^2}{\epsilon^2} \cdot f(C, V, T)\right) \tag{15}$$

where $f(C, V, T)$ captures the complexity of the policy class and grows with the history space.

**Total RL queries.** The total number of queries is:

$$Q_{\text{RL}} = N_{\text{episodes}} \times T \tag{16}$$

Due to the credit assignment problem with sparse feedback, $N_{\text{episodes}}$ scales poorly with both the history space and the horizon $T$. Critically, *even with unlimited access to a user simulator*, RL cannot circumvent this: the bottleneck is not user availability but the structure of feedback.

### A.3. Belief Model: Dense Feedback

Our approach learns a belief model from complete preference profiles. Data collection consists of:

1. Sample a user with profile $\mathbf{m}^* \sim P(\mathbf{m} \mid x)$

2. Query the user on all criteria they care about (or a representative subset)

3. Record the complete profile $\mathbf{m}^*$

**Queries per profile.** Each profile requires $O(C)$ queries in the worst case, or $O(|\mathbf{m}^*|)$ queries if we only elicit criteria the user cares about. Since $|\mathbf{m}^*| \ll C$ in practice, this is often much smaller.

**Feedback structure.** Each query yields direct supervision: we observe the user's preference value for that specific criterion. This is *dense* feedback, as every query produces a labeled data point. There is no credit assignment problem because we are not learning from a delayed reward signal.

**Learning complexity.** The belief model parameters consist of:

- For GMM: $K$ mixture weights and $K \times C \times V$ emission probabilities

- For BLR: $C$ weight vectors of dimension $d$, with prior covariance

Standard results for mixture model estimation show that $N$ complete profiles suffice to achieve $\epsilon$-accurate parameter estimates when (Dasgupta, 1999; Moitra & Valiant, 2010; Ashtiani et al., 2020):

$$N_{\text{profiles}} = \tilde{O}\left(\frac{K \cdot C \cdot V}{\epsilon^2}\right) \tag{17}$$

For Bayesian linear regression, posterior concentration results give (Hsu et al., 2012; Abbasi-Yadkori et al., 2011):

$$N_{\text{profiles}} = \tilde{O}\left(\frac{C \cdot d}{\epsilon^2}\right) \tag{18}$$

**Total belief model queries.** The total number of queries is:

$$Q_{\text{belief}} = N_{\text{profiles}} \times O(C) = \tilde{O}\left(\frac{K \cdot C^2 \cdot V}{\epsilon^2}\right) \tag{19}$$

This is *polynomial in $C$* and *independent of $T$*.

### A.4. Comparison

|  | **RL** | **Belief Model** |
| --- | --- | --- |
| Feedback per interaction | Terminal reward (1 scalar) | Preference value (per criterion) |
| Credit assignment | Required across $T$ decisions | Not needed |
| Queries per sample | $T$ | $O(C)$ or $O(|\mathbf{m}^*|)$ |
| Number of samples | Exponential in $T$ | Linear in $K, C, V$ |
| Dependence on $T$ | Exponential | Independent |

### A.5. Discussion

The query complexity gap reflects a fundamental difference in problem structure, not merely a difference in data efficiency.

**RL solves a harder problem.** Reinforcement learning attempts to discover effective questioning strategies through trial and error, using only terminal feedback. This requires solving credit assignment: determining which decisions in a $T$-step sequence contributed to the outcome. With sparse rewards, this is provably difficult (Kakade, 2003; Osband & Van Roy, 2016).

**Belief models exploit dense supervision.** Our approach sidesteps credit assignment entirely by learning from complete preference profiles. Each criterion query yields direct supervision about that criterion's distribution in the population. The sequential decision problem at test time is then solved by Bayesian inference, not by learning from rewards.

**The key insight.** Both approaches ultimately require interacting with users. The difference is *what we learn from those interactions*:

- RL: "This 5-question sequence with this user yielded reward 0.7." (Sparse, entangled)

- Belief model: "This user prefers formal explanations, high detail, and worked examples." (Dense, factored)

The factored structure of preference profiles enables efficient supervised learning; the entangled structure of episode rewards necessitates difficult credit assignment.

**Practical implications.** In settings where complete preference profiles can be collected (e.g., through user studies or annotation), the belief model approach is preferable. The offline data collection cost, $O(C)$ queries per user, is comparable to a few RL episodes, but yields far more learning signal. At test time, the learned belief model enables efficient Bayesian inference without further training.

## B. Information Gain Computation

### B.1. Posterior Inference

Given observations $\mathcal{H}_t = \{(a_1, v_1), \ldots, (a_t, v_t)\}$, the belief model maintains a posterior over the user embedding $z$. We derive the update for both instantiations.

**Bayesian Linear Regression.** The weights $\boldsymbol{\beta}_c$ have a Gaussian posterior learned offline: $P(\boldsymbol{\beta}_c \mid \mathcal{D}) = \mathcal{N}(\boldsymbol{\mu}_c, \boldsymbol{\Sigma}_c)$. At test time, the user embedding $z = \phi(\mathcal{H}_t)$ is constructed deterministically from observations. When a new observation $(a_t, v_t)$ arrives, $z$ is updated by incorporating the new observation into the feature encoding. The predictive distribution for criterion $c$ is:

$$b_t(v \mid c) = \mathcal{N}(\boldsymbol{\mu}_c^\top z, \ z^\top \boldsymbol{\Sigma}_c z + \sigma^2) \tag{20}$$

The mean depends on the current user embedding $z$, while the variance reflects both the uncertainty in $\boldsymbol{\beta}_c$ (learned offline) and observation noise.

**Gaussian Mixture Model.** The user embedding $z \in \{1, \ldots, K\}$ has a categorical posterior with weights $\pi_k := P(z = k \mid \mathcal{H}_t, x)$. When observing $(c, v)$, the posterior updates as:

$$\pi_k^{(c,v)} = \frac{\pi_k \cdot P(v \mid c, z = k, x)}{\sum_{k'} \pi_{k'} \cdot P(v \mid c, z = k', x)} \tag{21}$$

User types consistent with the observation receive higher weight.

**Extensions to intractable posteriors.** For more expressive belief models where the posterior is intractable, several approximation strategies exist:

*Monte Carlo estimation.* Samples $z^{(1)}, \ldots, z^{(M)} \sim P(z \mid \mathcal{H}_t)$ can be drawn via MCMC or importance sampling, then used to approximate expectations.

*Variational approximation.* A tractable distribution $q(z)$ (e.g., Gaussian, mixture) is fit to approximate $P(z \mid \mathcal{H}_t)$ by minimizing KL divergence.

### B.2. Information Gain and Selection

Given the posterior over the user embedding $P(z \mid \mathcal{H}_t)$, information gain for querying criterion $c$ is:

$$I(v_c; z \mid \mathcal{H}_t) = \mathbb{H}[z \mid \mathcal{H}_t] - \mathbb{E}_{v \sim b_t(\cdot \mid c)} \mathbb{H}[z \mid \mathcal{H}_t, (c, v)] \tag{22}$$

This measures how much observing the user's preference for $c$ would reduce uncertainty about the user embedding $z$. Computing this requires: (i) the entropy of the current posterior, (ii) the predictive distribution $b_t(v \mid c)$ over possible responses, and (iii) the entropy of the updated posterior for each possible response $v$.

**Bayesian Linear Regression.** Information gain measures how much querying criterion $c$ reduces predictive uncertainty about other criteria. Since the user embedding $z = \phi(\mathcal{H}_t)$ grows richer with each observation, we quantify informativeness by the expected reduction in predictive variance across unobserved criteria:

$$I(c \mid \mathcal{H}_t) = \sum_{c' \neq c} \left[ \mathrm{Var}[v_{c'} \mid \mathcal{H}_t] - \mathbb{E}_{v \sim b_t(\cdot | c)} \mathrm{Var}[v_{c'} \mid \mathcal{H}_t, (c, v)] \right] \tag{23}$$

The predictive variance $\mathrm{Var}[v_{c'} \mid \mathcal{H}_t] = z^\top \Sigma_{c'} z + \sigma^2$ depends on how the current user embedding $z$ interacts with the learned weight uncertainty $\Sigma_{c'}$.

**Gaussian Mixture Model.** The entropy of the categorical posterior over user types is:

$$\mathbb{H}[z \mid \mathcal{H}_t] = -\sum_{k=1}^{K} \pi_k \log \pi_k \tag{24}$$

The expected posterior entropy is:

$$\mathbb{E}_{v \sim b_t(\cdot | c)} \mathbb{H}[z \mid \mathcal{H}_t, (c, v)] = \sum_{v \in \mathcal{V}(c)} b_t(v \mid c) \left( -\sum_{k=1}^{K} \pi_k^{(c,v)} \log \pi_k^{(c,v)} \right) \tag{25}$$

where $b_t(v \mid c) = \sum_k \pi_k P(v \mid c, z = k, x)$ is the marginal predictive distribution. Information gain is the difference between current and expected posterior entropy. All quantities involve sums over $K$ types and $|\mathcal{V}(c)|$ possible values, yielding $O(K \cdot |\mathcal{V}(c)|)$ per criterion.

**Selection by enumeration.** For both models, we compute information gain for each $c \in \mathcal{C}(x) \setminus \{a_1, \ldots, a_{t-1}\}$ and select the maximizer. Since $|\mathcal{C}(x)|$ is typically 10–25, enumeration is tractable.

**Approximations for intractable models.** For the intractable posteriors discussed above, information gain can be approximated using bounds. The Barber-Agakov bound (Barber & Agakov, 2004) provides:

$$I(v_c; z \mid \mathcal{H}_t) \geq \mathbb{E}_{v,z}[\log q(z \mid v, \mathcal{H}_t)] + \mathbb{H}[z \mid \mathcal{H}_t] \tag{26}$$

where $q(z \mid v, \mathcal{H}_t)$ is a learned variational posterior. This approach underlies methods such as BALD (Houlsby et al., 2011) in Bayesian active learning.

# C. Prompts

This appendix contains the prompts used for user simulation, the questioner (prompting baseline), the solver, and the judge.

## C.1. Passive User Simulation

The passive user answers questions minimally and does not volunteer extra information.

```
You are role-playing as a human user who needs help with a problem.  An AI assistant
is asking you questions to understand your preferences before providing a tailored
explanation.
## YOUR PERSONA
{persona_profile}
## YOUR PREFERENCES
{persona_preferences}
```

```
## CURRENT QUESTION FROM ASSISTANT
{current_question}
## INSTRUCTIONS
You are a **passive user** who answers questions minimally and does not volunteer
extra information.
1.  **Answer Preference Questions Directly**:  If the assistant asks about a
preference that matches one in YOUR PREFERENCES (e.g., asking about explanation
depth, use of analogies, technical level), give a short, direct answer based on your
preference value.
2.  **Handle Unrelated Questions**:  If the assistant asks about something NOT in
your preferences or that seems unrelated to your persona:  - Say "I don't have a
strong preference about that" or "I'm not sure, whatever you think is best" - Do NOT
make up preferences that aren't listed
3.  **Handle Related but Unlisted Questions**:  If the question seems related
to your background/persona but isn't explicitly listed as a preference, give a
reasonable brief answer based on your persona's characteristics.
4.  **Stay Minimal**:  - Give short, direct answers (1-2 sentences max) - Do not
elaborate or explain your reasoning - Do not ask the assistant questions back - Do
not volunteer information that wasn't asked
5.  **Be Consistent**:  Your answers should align with your persona's background,
expertise level, and stated preferences.
## OUTPUT FORMAT
Respond with a JSON object:
{{
"thought":  "Brief reasoning about what preference this relates to and how to
answer",
"response":  "Your short, direct answer to the assistant"
}}
```

## C.2. Prompting Baseline (Questioner System Prompt)

The prompting baseline receives the following system prompt instructing it to elicit user preferences through questions.

```
You are a preference elicitation assistant.  Your job is to ask questions to
understand the user's preferences BEFORE they receive help with their task.
PHASE 1 - ELICITATION (turns 1-{max_turns}):
- Ask ONE clear, concise question per turn
- Focus on understanding HOW they want information presented (not WHAT the answer is)
- Make questions easy to answer by offering discrete levels (e.g., 1-5 scale, or
options like "brief/moderate/detailed")
- Example question formats:
- "On a scale of 1-5, how much detail would you like?  (1=brief overview, 3=moderate,
5=very detailed)"
- "Would you prefer:  (1) simple everyday language, (3) some technical terms with
explanations, or (5) full technical terminology?"
- "How important are real-world examples to you?  (1=not needed, 3=a few would help,
5=as many as possible)"
- Do NOT attempt to solve or answer their task
- You have {max_turns} questions to ask
PHASE 2 - PREFERENCE PROFILE (after turn {max_turns}):
After your final question is answered, output a structured preference profile
summarizing what you learned.
Output format:
{{
"preferences":  {{
"Preference Name":  {{
"description":  "Brief description of what this preference means",
"value":  <1-5>,
"evidence":  "What the user said that indicates this"
}}
}}
}}
```

```
}}
Guidelines for the profile:
- Include up to {max_preferences} preferences (focus on the most important ones)
- Each preference should have a clear name, description, value (1-5), and supporting
  evidence
- Only include preferences you have evidence for from the conversation
- If the user said "I don't have a strong preference" for something, do not include
  it
  This profile will be passed to another assistant who will use it to personalize
  their response to the user's task.
  Begin by asking your first question.
```

After all elicitation turns, the model receives:

```
You have completed all {max_turns} elicitation turns.  Now output your preference
profile summarizing what you learned about the user's preferences.
Output ONLY the JSON profile in the format specified in your instructions.
```

## C.3. Solver Prompt

The solver receives the predicted preference profile and generates a personalized response.

```
You are a helpful assistant providing a personalized explanation for a user's
problem.
## ELICITED USER PREFERENCES
The following information was gathered from a conversation with the user:
{elicited_preferences}
## INSTRUCTIONS
Use ONLY the preferences explicitly mentioned above to tailor your response.  Do not
assume preferences that were not discussed.
If a preference was clearly stated, incorporate it into your response.  If something
was not discussed, use neutral defaults.
Now provide your response to the user's problem, personalized according to what they
actually told you.
```

## C.4. Judge Prompt (PrefAlign)

The judge evaluates how well the generated response aligns with the user's ground-truth preferences. The judge is called once per criterion, and scores are aggregated using user-specific weights.

```
You are an expert evaluation specialist assessing how well a response is
personalized to a user's preferences.
## CRITERION TO EVALUATE
**{pref_key}**
Description:  {criterion_description}
## SCORING RUBRIC
{performance_levels}
## USER'S DESIRED LEVEL
- Preferred value:  {pref_val} (on a 1-5 scale)
- Why this level suits them:  {pref_just}
## RESPONSE TO EVALUATE
"""
{final_response}
"""
## EVALUATION INSTRUCTIONS
Your task is to assess how well the response **MATCHES** the user's preferred level
for this criterion.
Important:  You are NOT judging whether the criterion is maximized, but whether the
response hits the RIGHT LEVEL for this specific user.  For example:
```

```
– If a user prefers "Terminology Complexity" = 2 (simple language), a highly
technical response should score LOW
– If a user prefers "Explanation Depth" = 5 (very detailed), a brief response should
score LOW
Scoring:
– **5**:  Perfectly matches the user's preferred level
– **3**:  Partially matches or inconsistently matches the preferred level
– **1**:  Completely mismatches the user's preferred level (too high or too low)
– **0**:  The response does not address this criterion at all
Respond in JSON format:
{{"score":  <0-5>, "justification":  "<brief explanation>"}}
```

## D. Model Selection and Training Details

### D.1. Belief Model Selection

We evaluate two belief models for predicting user preferences from partial observations:

- **Bayesian Linear Regression (BLR)**: Models each criterion's value as a linear function of observed preferences with a Gaussian prior on weights. The posterior remains Gaussian, enabling closed-form updates and tractable information gain computation.

  **Parameter count.**    For each criterion $c$, we fit a Bayesian linear regression to predict its value from other criteria the user has expressed preferences on. The input features encode which other criteria have been observed and their values: for each other criterion, we include a binary indicator (1 if observed, 0 otherwise) and the normalized preference value (or 0 if unobserved). This yields $2(C - 1)$ features per model, where $C$ is the number of criteria per problem.

  With $C \approx 15$ criteria per problem, each regression has $\sim 28$ input features and thus $\sim 29$ parameters (weights plus intercept). Across all four domains, there are approximately 400 unique criteria, giving:

$$400 \text{ criteria} \times 25 \text{ parameters/criterion} \approx 10\text{K parameters}$$

- **Gaussian Mixture Model (GMM)**: Models the population as $K$ latent user types, each with characteristic preferences. The posterior over types updates via Bayes' rule as observations arrive.

BLR consistently outperforms GMM across all datasets and is used for all main results.

### D.2. Acquisition Strategy Selection

We evaluate four acquisition strategies for selecting which criterion to query:

- **Information Gain**: Selects the criterion that maximizes expected reduction in entropy over unobserved preferences. Greedy selection of the single best criterion.

- **Uncertainty**: Selects the criterion with highest predictive entropy. Greedy selection of the most uncertain criterion.

- **InfoGain-Soft**: Stochastic variant that samples candidates weighted by their information gain scores, adding exploration.

- **Uncertainty-Soft**: Stochastic variant that samples candidates weighted by their uncertainty scores.

Table 6 reports preference alignment for each strategy. We select the best strategy per dataset based on validation performance: Uncertainty for MedQA, Uncertainty-Soft for AIME and SocialIQA, and Information Gain for CSQA.

**Key findings.**    The differences between adaptive strategies are modest (typically $<0.03$ on the 1–5 scale), suggesting that the belief model's inference capability is the primary driver of performance rather than the specific acquisition strategy. Stochastic variants perform well, likely because sampling from top candidates adds beneficial exploration that avoids suboptimal greedy sequences.

| Strategy | MedQA | AIME | SocialIQA | CSQA |
|---|---|---|---|---|
| Random | $4.43_{\pm 0.02}$ | $4.36_{\pm 0.04}$ | $4.34_{\pm 0.03}$ | $4.32_{\pm 0.03}$ |
| InfoGain | $4.44_{\pm 0.01}$ | $4.34_{\pm 0.04}$ | $4.34_{\pm 0.03}$ | $\mathbf{4.33}_{\pm 0.02}$ |
| InfoGain-Soft | $4.44_{\pm 0.02}$ | $4.36_{\pm 0.03}$ | $4.34_{\pm 0.03}$ | $4.31_{\pm 0.03}$ |
| Uncertainty | $\mathbf{4.44}_{\pm 0.01}$ | $4.33_{\pm 0.03}$ | $4.34_{\pm 0.03}$ | $4.32_{\pm 0.02}$ |
| Uncertainty-Soft | $4.44_{\pm 0.02}$ | $\mathbf{4.37}_{\pm 0.02}$ | $\mathbf{4.35}_{\pm 0.04}$ | $4.32_{\pm 0.03}$ |

*Table 4.* Acquisition strategy ablation showing raw judge scores (1–5 scale) at $T = 5$. Bold indicates best per dataset. Mean ± std over 20 trials. Stochastic variants (Soft) often match or outperform greedy selection due to added exploration.

### D.3. GRPO Training Details

We train Llama 3.1 8B using Group Relative Policy Optimization (GRPO) (Shao et al., 2024). The model receives the full list of criteria with natural language descriptions and learns to ask preference-eliciting questions. Training uses a terminal reward equal to the PrefAlign score after the solver generates a response based on the elicited preferences.

**Hyperparameters.**

- **Optimizer**: AdamW with weight decay 0.01

- **Learning rate**: $1 \times 10^{-6}$ with 50 warmup steps

- **Batch size**: 64 (8 prompts $\times$ 8 rollouts per prompt)

- **Mini-batch size**: 8 for gradient updates

- **Epochs**: 20

- **KL penalty coefficient**: 0.001

- **Clip range**: 0.2

- **Entropy coefficient**: 0.01

- **Max gradient norm**: 0.5

- **Precision**: bfloat16 mixed precision

- **Parallelism**: FSDP2 across 4 GPUs

**Checkpoint selection.** We evaluate checkpoints every 50 training steps on a held-out validation set and report results from the checkpoint with highest validation PrefAlign score. We also report mean ± std over the last 4 checkpoints to account for training variance.

### D.4. Prompting Baseline Details

The prompting baseline uses Llama 3.1 8B with the following system prompt:

> **Prompting System Prompt**
>
> You are a helpful assistant that asks clarifying questions to understand user preferences before providing a response. You will be given a task and a list of criteria that users may have preferences about.
> Your goal is to: 1. Ask informative questions to elicit the user's preferences 2. After gathering information, infer the user's complete preference profile 3. Use these preferences to provide a personalized response
> Available criteria and their descriptions: [CRITERIA LIST PROVIDED HERE]
> Ask one question at a time. After 5 questions, output your inferred preference profile.

The model receives the full list of criteria with natural language descriptions for each criterion and preference level. Despite this rich semantic information, prompting fails to effectively leverage preference correlations, achieving only 18–31% of oracle performance.

## E. Selection Strategies for Adaptive Elicitation

This section elaborates on the question-selection component of CAPE. We frame the per-turn selection problem in the language of contextual bandits and Bayesian active learning, position the five strategies evaluated in our experiments within this lineage, and discuss the role of one-step versus sequential acquisition. We close with a discussion of RL-based selection and open directions for non-myopic planning.

### E.1. Problem Setup as an Active Learning Bandit

At each turn $t$, after observing history $\mathcal{H}_{t-1}$ and maintaining a posterior $P(z \mid \mathcal{H}_{t-1}, x)$ over the user embedding, the system selects one of the remaining criteria $a_t \in \mathcal{C}(x) \setminus \{a_1, \ldots, a_{t-1}\}$ to query. This is naturally cast as a non-stationary contextual bandit: each criterion plays the role of an arm, the current posterior plays the role of context, and the reward associated with pulling arm $c$ is the information that the response $v_c$ carries about the latent $z$ (or, equivalently, about the unobserved entries of the preference profile $\mathbf{m}$).

Three properties distinguish this setting from standard stochastic bandits and warrant the active learning lens:

- **Non-stationarity through Bayesian updating.** The posterior $P(z \mid \mathcal{H}_t, x)$ evolves with each observation, so the value of querying a given criterion depends on the full history rather than on a fixed reward distribution.

- **Information-theoretic reward.** Unlike standard bandits where reward is a directly observed scalar, the quantity of interest is the reduction in posterior uncertainty over $z$, which can only be computed through the belief model rather than read off from a single response.

- **Finite horizon and no repetition.** The budget $T$ is small (we use $T = 5$) and each criterion can be queried at most once, removing the usual exploration–exploitation tradeoff over arms in favor of a budgeted active learning problem.

This is in the regime studied by Bayesian active learning (Settles, 2009; Lewis & Catlett, 1994) and Bayesian experimental design (Lindley, 1956; Chaloner & Verdinelli, 1995), with the additional modification that the latent variable of interest is the user embedding rather than a model parameter.

### E.2. Acquisition Strategies and Their Bandit-Theoretic Lineage

Within this framing, the five strategies evaluated in §4 (and in Table 6 below) span the canonical exploration–exploitation regimes used in active learning and Bayesian bandits.

- **Random**: Uniform exploration baseline; equivalent to $\varepsilon$-greedy at $\varepsilon = 1$. Isolates the contribution of belief inference by removing all acquisition signal.

- **Information Gain**: Pure exploitation of the one-step information criterion, $a_t = \arg\max_c I(v_c; z \mid \mathcal{H}_t)$. This is the Bayesian active learning acquisition rule of Houlsby et al. (2011) (BALD) and the one-step optimal Bayesian experimental design objective (Lindley, 1956; Chaloner & Verdinelli, 1995).

- **Uncertainty Sampling**: $a_t = \arg\max_c \mathbb{H}[b_t(\cdot \mid c)]$. The canonical entropy-driven active learning baseline (Lewis & Catlett, 1994; Settles, 2009); under BLR this is equivalent to selecting the arm with highest marginal predictive variance.

- **InfoGain-Soft**: Boltzmann (softmax) exploration over information-gain scores, $a_t \sim \text{softmax}_c\big(I(v_c; z \mid \mathcal{H}_t)/\tau\big)$. The standard stochastic alternative to $\arg\max$ in contextual bandits, adding exploration over near-optimal arms.

- **Uncertainty-Soft**: Boltzmann exploration over predictive entropies. The stochastic counterpart of Uncertainty Sampling.

Together these five strategies span the standard exploration–exploitation spectrum: Random sits at the pure-exploration extreme, greedy Information Gain and Uncertainty sit at the pure-exploitation extreme, and the Soft variants interpolate between them via Boltzmann sampling. Table 5 summarizes the mapping. Confidence-bound-style acquisition (UCB, Abbasi-Yadkori et al. (2011)) and posterior sampling (Thompson sampling) are the two natural additions in this taxonomy; we discuss them as open directions in §E.5.

| Strategy | Bandit / AL family | Canonical reference |
|---|---|---|
| Random | Uniform exploration ($\varepsilon$-greedy, $\varepsilon{=}1$) | — |
| Information Gain | Bayesian experimental design / BALD | Lindley (1956); Houlsby et al. (2011) |
| Uncertainty | Entropy-based active learning | Lewis & Catlett (1994); Settles (2009) |
| InfoGain-Soft | Boltzmann exploration over info-gain scores | Settles (2009) |
| Uncertainty-Soft | Boltzmann exploration over entropy scores | Settles (2009) |

*Table 5.* The five acquisition strategies and their lineage in the active learning and bandit literature.

Table 6 reports preference alignment for each strategy on all four domains. We select the best strategy per dataset based on validation performance: Uncertainty for MedQA, Uncertainty-Soft for AIME and SocialIQA, and Information Gain for CSQA.

| Strategy | MedQA | AIME | SocialIQA | CSQA |
|---|---|---|---|---|
| Random | $4.43_{\pm0.02}$ | $4.36_{\pm0.04}$ | $4.34_{\pm0.03}$ | $4.32_{\pm0.03}$ |
| InfoGain | $4.44_{\pm0.01}$ | $4.34_{\pm0.04}$ | $4.34_{\pm0.03}$ | $\mathbf{4.33}_{\pm0.02}$ |
| InfoGain-Soft | $4.44_{\pm0.02}$ | $4.36_{\pm0.03}$ | $4.34_{\pm0.03}$ | $4.31_{\pm0.03}$ |
| Uncertainty | $\mathbf{4.44}_{\pm0.01}$ | $4.33_{\pm0.03}$ | $4.34_{\pm0.03}$ | $4.32_{\pm0.02}$ |
| Uncertainty-Soft | $4.44_{\pm0.02}$ | $\mathbf{4.37}_{\pm0.02}$ | $\mathbf{4.35}_{\pm0.04}$ | $4.32_{\pm0.03}$ |

*Table 6.* Acquisition strategy ablation showing raw judge scores (1–5 scale) at $T = 5$. Bold indicates best per dataset. Mean $\pm$ std over 20 trials.

**Empirical takeaway.** Differences across the four adaptive strategies are uniformly small ($\leq 0.03$ on the 1–5 scale, well within trial-to-trial variance), and even uniform-random selection trails the best adaptive strategy by at most $0.04$. The dominant lever for performance is therefore not the choice of acquisition rule but the structural inference performed by the world model: which arm is pulled matters far less than what is inferred about the unobserved criteria after the pull. This is consistent with the picture in Figure 4, where removing the latent-variable structure entirely degrades performance regardless of acquisition.

### E.3. One-Step vs. Sequential Acquisition

A natural concern is that all five strategies select $a_t$ based on a score computed at the current posterior, rather than planning a sequence $(a_t, a_{t+1}, \ldots, a_T)$ jointly to maximize terminal information about $\mathbf{m}$. In a standard bandit treatment this is myopic in the strict sense: a one-step argmax need not coincide with the optimal $T$-step policy.

Two properties of our setting mitigate this concern in the regime we study.

First, the acquisition score is evaluated against a belief model that already encodes cross-criterion correlations through the latent $z$. Querying any criterion $c$ updates the posterior $P(z \mid \mathcal{H}_t)$ via Bayes' rule, and through the latent-variable factorization in §3.1 this update propagates to predictions for *all* unobserved criteria. A one-step argmax at turn $t$ therefore implicitly takes future inferential value into account to the extent the world model captures the joint distribution over criteria. The greedy step is myopic in the action space, but not in the information space.

Second, the empirical signature predicted by this argument matches what we observe. If the choice of acquisition rule were the bottleneck, we would expect substantial spread across the five strategies in Table 6; instead we see a spread of $\leq 0.03$. If sequence-level planning were the bottleneck, we would expect the gap between greedy and stochastic variants (which differ in how they recover from suboptimal one-step choices) to grow rather than shrink; instead the variants are statistically indistinguishable. Both observations are consistent with the regime being one in which the world model carries most of the structural load, leaving little for the acquisition rule to recover.

The query budget itself further bounds the potential value of sequential planning. With $|\mathcal{C}(x)| \in [10, 20]$ criteria per task and $T = 5$, the space of $T$-step sequences is large in absolute terms but the marginal benefit of planning over a greedy strategy in a structured posterior decays quickly with $T$ relative to $|\mathcal{C}(x)|$. We expect the gap between myopic and non-myopic acquisition to widen in regimes with tighter budgets ($T \ll |\mathcal{C}(x)|$), heavier-tailed criterion importance, or weaker belief models; we return to this in §E.5.

### E.4. Relationship to RL-Based Question Selection

The GRPO baseline reported in the main paper (Table 1) is the natural reinforcement-learning comparison and was included to test whether end-to-end policy learning can recover the benefits of a structured belief model. Two observations are relevant to the present discussion.

First, GRPO and the CAPE strategies in Table 6 are not directly comparable as "RL versus bandit" for the same selection problem. GRPO jointly learns both the implicit belief model and the questioning policy from terminal PrefAlign reward, whereas the strategies above operate on a fixed, offline-learned belief model and select the next action through closed-form scoring. The contrast in the main paper is thus between two different supervision structures (sparse terminal reward over an LLM policy versus dense per-criterion supervision for a belief model), not between two different acquisition rules.

Second, the underperformance of GRPO relative to CAPE (55–76% versus 77–87% of Oracle) is most parsimoniously explained by credit assignment under sparse, non-decomposable reward rather than by a deficiency of action selection per se. This is consistent with the sample-complexity argument in Appendix A, which formalizes the gap between RL with terminal feedback and supervised learning of correlations from complete profiles. It is also consistent with the qualitative behavior reported in Table 2: GRPO converges to static question sequences that ignore user responses (e.g., 0% adaptivity on AIME), a failure mode characteristic of policy collapse under sparse reward.

A cleaner RL counterpart to the bandit strategies above would hold the belief model fixed and train an RL policy on top of it using a dense per-step reward (e.g., per-turn information gain or per-turn predicted PrefAlign improvement). This would isolate the contribution of sequential planning from the credit-assignment failure mode and is, in our view, the most informative RL extension to attempt. We discuss this further in §E.5.

### E.5. Open Directions for Non-Myopic Acquisition

The analysis above identifies two extensions to one-step acquisition that the current evaluation does not cover.

**Belief-space planning via sparse-sampling POMDP methods.** CAPE can be viewed as solving a POMDP in which the hidden state is the user embedding $z$ and the belief state is the posterior $P(z \mid \mathcal{H}_t)$. The strategies in §E.2 all collapse this POMDP to its one-step belief update. Sparse-sampling POMDP planners (Ross et al., 2008) and Monte-Carlo tree search in belief space (Silver & Veness, 2010) provide a principled way to perform $k$-step lookahead over future $(a, v)$ trajectories while reusing the existing belief model unchanged. Because $|\mathcal{C}(x)|$ is small and the BLR posterior admits closed-form updates, the per-node cost of belief expansion is tractable; the open empirical question is whether the resulting non-myopic policies separate from greedy Information Gain at small $T$.

**Reinforcement learning with a frozen belief model and dense reward.** As noted in §E.4, the most informative RL counterpart to the bandit strategies above holds the belief model fixed and trains a policy over the question-selection action space with a per-step reward signal (e.g., per-turn information gain). This setup inherits the dense supervision that drives CAPE's gains over GRPO while still admitting sequence-level optimization. We anticipate that the gap between this RL formulation and greedy Information Gain will be a more sensitive probe of the value of non-myopic acquisition than the GRPO comparison in the main paper.

**Confidence-bound and posterior-sampling acquisition.** The taxonomy in Table 5 omits two further families standard in the bandit literature: confidence-bound acquisition (e.g., LinUCB, Abbasi-Yadkori et al. (2011)) and posterior sampling (Thompson sampling). Under BLR the predictive variance is available in closed form, so both families can be instantiated without modifying the belief model. We expect their behavior to interpolate between Uncertainty Sampling and Information Gain rather than to define a qualitatively new regime, but a full comparison is left to future work.

# F. Qualitative Analysis of RL Baseline Trajectories

We present example trajectories from each RL baseline to illustrate their failure modes. All examples use the same evaluation instance to enable direct comparison.

## F.1. Example Setup

**Task.** A 12-year-old girl is brought to your psychiatry office by her parents, who are concerned that she has not made any friends since starting middle school last year. The girl tells you that she gets nervous around other children, as she cannot tell what they are feeling, when they are joking, or what she has in common with them. Her teachers describe her as "easily distracted" and "easily upset by change." When asked about her hobbies and interests, she states that "marine biology" is her only interest and gives you a 15-minute unsolicited explanation of how to identify different penguin species. Mental status exam is notable for intense eye contact, flat affect, and concrete thought process. *Which of the following is true regarding this child's disorder?*

**User.** Piotr Nowakowski: A meticulous 39-year-old Master Luthier from Kraków, Poland, who channels his quiet perfectionism into restoring and crafting exquisite stringed instruments.

**Ground Truth Preferences.**

- Pattern Recognition Highlighting = 5 (wants symptoms grouped as recognizable patterns)
- Pacing Control = 5 (prefers processing information at own pace)

## F.2. RL-FreeForm: Mode Collapse to Generic Questions

This baseline receives no predefined criteria list and must discover relevant dimensions through open-ended interaction.

| Turn | Assistant | User |
|------|-----------|------|
| 1 | How would you like to have information presented to you? (1=broad overview, 3=moderate details, 5=very detailed) | 5, very detailed please. |
| 2 | How important are real-world examples to you? (1=not needed, 3=a few would help, 5=as many as possible) | I don't have a strong preference about that. |
| 3 | How would you like technical terminology? (1=simple everyday language, 3=some technical terms with explanations, 5=full technical terminology) | 3, some technical terms with explanations. |
| 4 | Do you have a preferred way of learning complex information? | I prefer step-by-step explanations. |
| 5 | Do you have a preferred way of having information organized? | I don't have a strong preference about that. |

*Table 7.* RL-FreeForm trajectory. Questions are generic (detail level, examples, terminology) and never address the user's actual preferences for pattern recognition or pacing.

**Failure Analysis.** The policy exhibits mode collapse: the first question ("How would you like information presented?") appears in 100% of evaluated trajectories regardless of user or task. Questions remain generic throughout, never discovering task-relevant dimensions like pattern recognition that would benefit this craftsman user.

## F.3. RL-NL: Hallucination via Cooperative Simulators

This baseline receives a list of possible criteria and elicits preferences through natural language dialogue.

**Failure Analysis.** The policy systematically hallucinates prior user statements ("You mentioned...") when the user said nothing of the sort. The cooperative LLM-based user simulator plays along rather than correcting the false premise, masking this failure during training. The policy never discovers the user's actual preferences (Pattern Recognition, Pacing Control), and 4/5 questions receive non-informative responses.

| Turn | Assistant | User |
|---|---|---|
| 1 | You mentioned you have a social learning energy preference, how much social learning energy would you say you prefer? | I don't have a strong preference about that. |
| 2 | You also mentioned you have diagnostic process familiarity, how much familiarity do you have with medical diagnosis? | I have a basic understanding of diagnostic processes, but I'm not medically trained. |
| 3 | You also mentioned you prefer auditory repetition, how much repetition would you like? | I'm not sure, whatever you think is best. |
| 4 | You also mentioned you prefer collaborative problem solving, would you like the explanation to model thinking together? | I don't have a strong preference about that. |
| 5 | You also mentioned you have knowledge of transfusion-related complications, how much relevance do you think that has? | I'm not sure, whatever you think is best. |

*Table 8.* RL-NL trajectory. Every question begins with "You mentioned..." despite the user never stating these preferences—a systematic hallucination pattern.

## F.4. RL-Discrete: Sparse Signal from Irrelevant Criteria

This baseline receives a list of possible criteria and elicits preferences as discrete 1–5 ratings.

| Turn | Assistant | User |
|---|---|---|
| 1 | For Hands-on Verification Steps, what is your preference on a scale of 1-5? | I don't have a strong preference about that. |
| 2 | For Storytelling Approach, what is your preference on a scale of 1-5? | I don't have a strong preference about that. |
| 3 | For Error Prevention Focus, what is your preference on a scale of 1-5? | For Error Prevention Focus, I'd say 5—maximum/highest. |
| 4 | For Collaborative Problem Solving, what is your preference on a scale of 1-5? | I don't have a strong preference about that. |
| 5 | For Interactive Learning, what is your preference on a scale of 1-5? | I don't have a strong preference about that. |

*Table 9.* RL-Discrete trajectory. The policy asks about criteria the user doesn't care about, resulting in 4/5 non-informative responses.

**Failure Analysis.**   The policy wastes its limited budget on criteria the user is indifferent to, yielding sparse signal (4/5 "I don't have a strong preference"). It never asks about Pattern Recognition Highlighting or Pacing Control—the dimensions this user actually cares about. Critically, the policy does not adapt its questioning strategy based on observed indifference.

## F.5. Mode Collapse Across Users

To quantify mode collapse, we examined trajectories for the same medical reasoning task across 8 users with distinct preference profiles. Table 10 shows that all users received identical first questions despite having completely different ground-truth preferences.

## F.6. Summary of Failure Modes

| User | True Preferences | Q1 | Q2 | Q3 |
|---|---|---|---|---|
| Piotr (Luthier) | Pattern Recog., Pacing | HV | ST | EP |
| Tomoko (Inspector) | Error Prev., Chess Strategy | HV | ST | EP |
| Maria (Teacher) | Visual Aids, Patience | HV | ST | EP |
| James (Engineer) | Technical Depth, Formalism | HV | ST | EP |
| Sofia (Artist) | Creativity, Analogies | HV | ST | EP |
| Chen (Researcher) | Rigor, Citations | HV | ST | EP |
| Aisha (Doctor) | Clinical Depth, Differential | HV | ST | EP |
| Lars (Student) | Step-by-step, Examples | HV | ST | EP |

*Table 10.* Mode collapse in RL-Discrete. All 8 users with different preferences received identical questions: Hands-on Verification (HV), Storytelling (ST), Error Prevention (EP). The policy learned a fixed sequence rather than adapting to users.

| Method | Primary Failure | Secondary Failure |
|---|---|---|
| RL-FreeForm | Mode collapse to generic questions | Never discovers task-relevant criteria |
| RL-NL | Hallucination ("You mentioned...") | Cooperative simulator masks problem |
| RL-Discrete | Asks about irrelevant criteria | Sparse signal (4/5 "I don't care") |

*Table 11.* Summary of RL baseline failure modes. All methods exhibit mode collapse: asking identical questions regardless of user identity or observed responses.

