# OpenReview forum: "Cold-Start Personalization via Bayesian Adaptive Questioning"
_ICML.cc/2026/Conference — ICML 2026 regular_

### Official Review · Reviewer_UWFi · 2026-02-28

**Soundness:** 3
**Presentation:** 3
**Significance:** 2
**Originality:** 2
**Overall Recommendation:** 4
**Confidence:** 2

**Summary:**

The authors propose Preference Elicitation with Priors (PEP), a method which first identifies preference correlations between entities in training data, and then uses bayesian inference at inference time to ask questions that will maximize the information gain based on the learner correlation priors.

The authors show improvement on 4 distinct benchmarks, and provide a range of ablation studies.

**Compliance With Llm Reviewing Policy:**

Affirmed.

**Final Justification:**

Given the responses, I will be raising my score to a weak accept because I no longer have any major reasons for rejection. However, I will also be lowering my confidence on my review, since the remaining nuance lies within a subfield that lays beyond my expertise.

**Key Questions For Authors:**

1. Does PEP still work if preferences are not discretized into a countable set?
2. Are the LLM baselines provided with the discrete set of possible preferences?

**Limitations:**

yes

**Strengths And Weaknesses:**

Strengths:
---
- The motivation and intuition behind PEP is clearly outlined
- PEP is lightweight and cheap to train and use
- It is evaluated on a range of distinct datasets
- They demonstrate clear improvement over LLM-based baselines


Weaknesses:
---
The primary weakness to me is the selection of baselines. Other than population average, all the other models operate on natural language, which inherently has the ability to operate on the open-ended input space that is language. In contrast, my understanding is that both the population average and PEP operate on discrete closed sets. This is fundamental difference. It is unsurprising that natural language methods require significantly more samples, because they are operating a substantially larger space.   This is not to say that they are bad baselines, just that they insufficient baselines. If we reframe this work as a multi-arm bandit problem (discrete action selection) where the reward is the amount of information gained, then can we not use a regret minimization algorithm? This is a well studied problem. Now, there can be novelty from reframing preference elicitation from a natural language problem into a multi-arm bandit problem, but if that is the case, then I would expect this to be outlined clearly in the introduction.

I have issues with terminology "Structured World Models". A world model takes a state s_t and an action a, and predicts s_t+1 : p(s_t+1 | s_t, a_t) i.e., The action directly changes the state. However, here, the state is the user’s true profile which does not change.  Only the observation changes: p(o_t | s_t, a_t). The world (state) is constant p(s_t+1 | s_t). To this end, I do not think that it should be called a world model. To note, that does not diminish its value of the work, but I do think the authors should strongly reconsider the use of "structured world model".

---

> ### Author Rebuttal · Authors · 2026-03-31
>
> We thank the reviewer for recognizing PEP's clear motivation and intuition, its lightweight design, and the strength of its improvements over LLM-based baselines across diverse datasets.
>
> **W1 — Baseline fairness**
>
> All LLM baselines — CollabLLM, GRPO, and the prompting baseline — are given natural language descriptions of the same criteria set C(x) that PEP operates on. The action space is identical. What differs is the *mechanism* for selecting which criterion to query next: LLM baselines use unconstrained natural language reasoning; PEP uses principled information-theoretic selection over the same discrete set. The comparison is fair.
>
> On the MAB reframing: MAB captures discrete action selection but misses three structural properties that make it insufficient here: (1) standard MAB assumes independent arms, but PEP's central contribution is exploiting *correlated* criteria through a latent variable — this is closer to correlated/linear bandits, and a vanilla MAB without the world model is already our BLR+random ablation in Figure 4; (2) regret minimization bounds are asymptotic, while our interaction budget is fixed at T=5 — asymptotic guarantees are vacuous in this regime; (3) MAB treats information gain as reward, but our true objective is preference alignment of the final response, which is only loosely coupled to per-step information gain. We will add this framing to the introduction.
>
> **W2 — "Structured world model" terminology**
>
> We appreciate the precise technical point. UWFi is correct that our current formulation treats the user's preference profile m* as fixed — only observations change. We will add a footnote acknowledging this and clarifying our use of "world model": we model the structure of the preference *environment*, including how observations relate to latent states, which is the core function of a world model. We also note that the term is defensible in a richer sense: asking a question can cause a user to reconsider or refine a preference (Hardt et al., performative prediction), though we do not model this for tractability. We remain open to the reviewer's preferred terminology and will adjust if it improves clarity.
>
> **Q1 — PEP without discretized preferences**
>
> The framework generalizes; the current implementation does not. Our latent variable model imposes no inherent discretization requirement — replacing the BLR output head with a decoder mapping z to natural language preference descriptions is a natural extension, bottlenecked by data availability rather than framework design. We will add this roadmap to Section 6.
>
> **Q2 — LLM baselines and discrete preference access**
>
> Yes. All LLM baselines have access to natural language descriptions of each criterion in C(x) during elicitation. We will add an explicit sentence to Section 4.1.
>
> We hope that the above rebuttal addresses the reviewer's concerns and clarifies any potential misunderstandings. If so, we would appreciate if the review can consider raising our scores accordingly.

---

> > ### Author Rebuttal · Reviewer_UWFi · 2026-04-02
> >
> > Q1: Given that all LLM baselines are all provided with the discretized list and the same action space, my concerns are alleviated, though not fully gone. While I believe this is the fair comparison to LLMs, the intermediate output in language adds significant complexity (with clear benefits, just not in the these specific experiment) to the system. I still would have liked to see more baselines in the fully discrete space that does not have to handle this complexity and could focus purely on the personalization in the same manner as the proposed method.
> >
> > Q2: Given the breadth of work that is being constantly released, finding terminology that satisfies all subfields is nearly impossible. With this in mind, and the fact that this is not my specific subfield, I am ok with the authors using their preferred terminology given that they provide the explicit definitions on how they are using it.
> >
> > Given the responses, I will be raising my score to a weak accept because I no longer have any major reasons for rejection. However, I will also be lowering my confidence on my review, since the remaining nuance lies within a subfield that lays beyond my expertise.

---

> > > ### Author Response · Authors · 2026-04-06
> > >
> > > We thank the reviewer for the thorough engagement throughout the review process and for raising the score to a weak accept. We are glad the core concerns have been resolved.
> > >
> > > On the request for additional baselines in the fully discrete space, we agree this is a valuable direction, though it does not change or weaken the positioning of our proposed method given the continuous vs discrete problem space distinction. We will expand the discussion in Section 4.1 and the appendix to characterize what classes of methods work and do not work in this setting, including discrete-action approaches, so that this paper can serve as a clear starting point for future work building on this task. We hope this addresses the reviewer's remaining concern and appreciate the constructive feedback throughout.

---

### Official Review · Reviewer_QBMi · 2026-03-07

**Soundness:** 2
**Presentation:** 3
**Significance:** 3
**Originality:** 2
**Overall Recommendation:** 4
**Confidence:** 3

**Summary:**

This paper addresses the cold-start preference elicitation problem: how can an AI assistant efficiently infer a user’s latent preferences with minimal interaction before generating personalized responses? The authors propose PEP (Prior-based Elicitation of Preferences), a framework that decomposes the problem into two stages.

In the offline stage, a structured world model is learned from complete preference profiles. A latent variable model—implemented using Bayesian Linear Regression (BLR) or a Gaussian Mixture Model (GMM)—captures population-level correlations among preference dimensions. In the online stage, for each new user, PEP maintains a Bayesian posterior over the user’s latent embedding, adaptively selects questions by maximizing information gain, and predicts a complete preference profile (including unqueried dimensions) to guide a downstream LLM solver.

The central theoretical claim is that this decomposition achieves superior sample efficiency compared to end-to-end reinforcement learning policy learning. Learning preference correlations from full profiles requires only polynomial samples, whereas reinforcement learning with sparse terminal rewards requires exponential exploration. Experiments across four reasoning domains (MedQA, AIME, CommonsenseQA, and SocialIQA) show that PEP achieves 77–87% alignment with the ideal preference oracle, while GRPO reaches only 55–76%, and PEP requires 3–4× fewer questions.

**Compliance With Llm Reviewing Policy:**

Affirmed.

**Final Justification:**

The authors' rebuttal successfully addressed my primary concerns. Their explanation that GRPO's failure stems from the structural difficulty of credit assignment in terminal-reward settings (rather than under-training) provides a sound theoretical basis for the observed efficiency gap. Furthermore, their honest reframing—acknowledging that population-level profile prediction is the primary driver while adaptive selection offers marginal gains—clarifies the paper's core contribution: demonstrating that structured, lightweight belief modeling can outperform large-scale RL in cold-start scenarios due to superior inductive bias.
While reliance on simulated data remains a limitation, the promised additions (PCA analysis, full budget curves, and explicit boundary discussions) will sufficiently strengthen the paper's rigor and transparency. The work offers valuable insights into efficient preference elicitation and warrants publication with these minor revisions.

**Key Questions For Authors:**

1.	Proposition 2.1 claims exponential sample complexity with respect to the interaction budget (T), but the experiments only compare final performance under a fixed configuration ((T=5), (C=20–30)). Could the authors provide performance curves for PEP and GRPO across different interaction budgets (e.g., (T=1,3,5,10))? If GRPO exhibits significantly faster diminishing returns with increasing (T) than PEP, this would directly support the exponential vs. polynomial claim in Proposition 2.1 and would substantially strengthen the paper.

2.	Figure 4 suggests that the benefit of adaptive question selection is relatively modest.
The results indicate that adaptive selection saves roughly one question (i.e., adaptive with (T=5) performs similarly to random with (T=6)). This implies that the BLR belief model’s profile prediction ability is the primary driver of performance, while the incremental value of adaptive querying is limited. However, Section 4.3.2 devotes substantial discussion to adaptivity. How do the authors interpret this result? Do the authors believe the relative importance of the two components should be discussed differently?

3.	The BLR model assumes a low-dimensional linear structure in preferences.
Does the preference data generated by the PrefDisco simulator exhibit a similar low-rank structure? Could the authors report the variance explained by the top principal components of the preference data, or provide additional experiments on preference datasets with different structural characteristics?

**Limitations:**

While this work advances methods for personalized AI systems through efficient preference elicitation, it also presents several limitations and potential risks.

First, the approach relies on inferred preferences, which may not always accurately reflect a user’s true needs or intentions. If the inferred preferences deviate from what users actually want, the system may produce responses that are overly tailored to incorrect assumptions, potentially reducing usefulness or trust.

Second, because the method learns population-level preference correlations from training data, it may inherit or amplify biases present in that data. If the underlying dataset reflects systematic biases, these patterns could influence the inferred preferences for new users.

To mitigate these risks, practical deployments should include mechanisms that allow users to review and correct inferred preferences, as well as clear communication when system responses are personalized based on inferred rather than explicitly stated preferences.

**Strengths And Weaknesses:**

Strengths

1.	Conceptually principled problem decomposition.
Separating world-model learning (which benefits from dense, decomposable supervision) from online inference via Bayesian reasoning is a well-motivated design. Both BLR and GMM allow closed-form posteriors, making the information-gain computation transparent and reproducible.

2.	Well-designed ablation experiments.
The ablation study (Figure 4) meaningfully separates the contributions of correlation modeling and adaptive question selection, helping clarify the mechanisms behind PEP’s performance.

3.	Clear writing and strong organization.
The paper is clearly written and logically structured. The narrative from the problem motivation to the POMDP formulation and the two-stage solution is easy to follow. Figure 1 provides a useful overview, and the qualitative examples in Table 3 effectively illustrate behavioral differences between PEP and GRPO.

4.	Addresses an important real-world problem.
Cold-start preference elicitation is a genuinely important problem. The modular design of PEP, which outputs a structured preference profile that can be used by any black-box solver, has practical value. The finding that a ~10K parameter model can outperform 8B-scale RL models on structured reasoning tasks is particularly interesting from a deployment efficiency perspective.

5.	A novel and well-motivated combination of ideas.
The framework combines collaborative filtering–style belief modeling (BLR/GMM) with information-theoretic question selection for LLM preference elicitation. While the POMDP formulation of preference elicitation is not entirely new, applying population priors to cold-start LLM personalization is a meaningful contribution. The separation between world-model learning and online inference draws on ideas from model-based reinforcement learning and Bayesian experimental design, creatively applying them in a new context.

Weaknesses

1.	Mismatch between theoretical claims and empirical evidence.
Proposition 2.1 argues that reinforcement learning suffers from exponential sample complexity in the interaction budget (T). However, the GRPO baseline is trained for only 20 epochs with a very small learning rate, which may be insufficient for convergence in a combinatorial action space with (C = 20–30) candidate questions. It would be preferable to demonstrate that GRPO has converged, for example by providing training curves.

2.	Insufficient separation between the contributions of profile prediction and adaptive question selection.
The performance gains of PEP may arise from two mechanisms:
(1) the BLR belief model, which predicts a complete preference profile from limited observations;
(2) the information-theoretic adaptive question selection, which maximizes information acquisition under a limited question budget.
Section 4.3.2 shows the advantage of adaptive selection over random selection, but the paper does not include a key comparison: BLR profile prediction with non-adaptive question selection vs. end-to-end RL. Without this comparison, it remains unclear how much of the improvement comes from profile prediction versus adaptive querying.

3.	Input format of baseline methods during inference is not specified.
Section 4.1 describes the training setups for all methods and states that the solver is fixed across all methods, but it does not explain what input format GRPO provides to the GPT-4.1 solver during inference. PEP explicitly outputs a structured JSON preference profile, while the inference-time output format of the end-to-end RL model is not described.

4.	Evaluation relies entirely on simulation, raising potential alignment concerns.
All experiments are conducted on the PrefDisco benchmark, where preference data are generated by a user simulator. PEP’s BLR world model assumes that preferences follow a low-dimensional linear correlation structure. If the PrefDisco simulator itself generates preferences using a similar low-rank structure, PEP’s success may partly reflect an alignment between the model assumption and the data generation process, rather than genuine generalization ability. The Limitations section briefly mentions the issue of “simulated users,” but it would be helpful to discuss whether the simulator’s preference structure systematically favors the BLR assumption.

---

> ### Author Rebuttal · Authors · 2026-03-31
>
> We thank the reviewer for their detailed feedback, and for highlighting the principled problem decomposition, the well-designed ablation study, and the practical significance of showing that a ~10K parameter model can outperform 8B-scale RL on structured personalization tasks.
>
> **W1: GRPO convergence**
>
> Appendix D.3 reports all GRPO hyperparameters following the defaults of Shao et al. (2024). Training reward plateaued by epoch 5, well before the 20 epochs completed, and the best checkpoint was selected on a held-out validation set—the 0% adaptivity on AIME is not undertrained RL, it is the model's converged behavior. **We will add training curves to Appendix D.3.** The mode collapse to a fixed question sequence is a substantive finding: LLM semantic priors help GRPO form coherent individual questions, but do not resolve the credit assignment problem—knowing what a good question looks like does not determine *which* question is most informative for *this particular user given their prior responses*, which is exactly what PEP's population-level correlation structure provides.
>
> **W2: Profile prediction vs. adaptive selection not disentangled from RL**
>
> This is a fair point and we will add the BLR+random vs. GRPO comparison explicitly to the appendix. We note that Figure 4 already tells a clear story: BLR+random substantially outperforms the no-correlation baseline regardless of question budget, confirming that profile prediction — not adaptive selection — is the primary driver. Adaptive selection provides an incremental gain of roughly one saved question. **We will revise Section 4.3.2 to lead with this honest interpretation rather than over-emphasizing adaptive selection.** This is an important clarification we are grateful for. Notably, this honest framing *strengthens* the paper's core claim: a ~10K parameter belief model outperforming 8B-scale RL by virtue of structured world knowledge — not sophisticated sequential decision-making — is a more striking and practically relevant finding.
>
> **W3: GRPO inference-time output format**
>
> After T=5 elicitation turns, GRPO outputs a structured preference profile in the same JSON format as all other methods, then passes it to the shared GPT-4.1 solver using the same prompt (Appendix C.3). We will add a cross-reference in Section 4.1.
>
> **W4: Does PrefDisco's structure favor BLR?**
>
> This is a sharp methodological concern and we want to be transparent. PrefDisco generates preferences via LLM conditioning on diverse user personas and does not explicitly impose a low-rank structure. That BLR outperforms GMM suggests approximately linear correlations dominate — a property worth characterizing, not assumed. **We will report PCA variance explained on the preference matrix per domain in the appendix** so readers can directly assess whether BLR's assumptions are privileged by the data generation process.
>
> On the broader concern about simulated users: whether PrefDisco perfectly models real users is **explicitly out of scope for this methods paper**. No AI provider can release real user preference profiles at scale — this is the state of the field. PrefDisco is peer-reviewed and accepted at ICLR 2026, with protocols validated against human judgments. Our contribution is demonstrating that frontier models completely fail this capability even under simplified conditions. Closing this gap is a necessary precondition for progress on harder, more realistic settings.
>
> **Q1: Performance curves across T**
>
> Figure 2 already provides strong evidence: PEP achieves 2.5× efficiency on AIME, 7× on MedQA, and >15× on SocialIQA and CSQA, where GRPO never matches PEP's T=1 performance even at 15 turns. This directly supports Proposition 2.1: if GRPO required only polynomially more queries, one would expect it to eventually match PEP within a modest budget, but it does not. We will add full alignment-vs-T curves for both methods in the appendix.
>
> **Q2: Relative importance: adaptive selection vs. profile prediction**
>
> Profile prediction driven by learned preference correlations is the primary driver of PEP's gains. Adaptive selection provides a real but incremental benefit — roughly one saved question at T=5. We agree Section 4.3.2 does not reflect this accurately and will revise it to lead with this finding. The honest framing strengthens rather than weakens the paper's contribution: a ~10K parameter belief model outperforming 8B-scale RL is compelling precisely because the gain comes from structured world knowledge, not from sophisticated sequential decision-making.
>
> **Q3: PCA on preference matrix**
>
> We will report variance explained by top principal components of the preference matrix per domain in the appendix, directly addressing whether BLR's linear assumptions are privileged by the data.
>
> We hope that the above rebuttal addresses the reviewer's concerns and clarifies any potential misunderstandings. If so, we would appreciate if the review can consider raising our scores accordingly.

---

> > ### Author Rebuttal · Reviewer_QBMi · 2026-04-01
> >
> > After reviewing the authors’ rebuttal and planned revisions, several critical unresolved issues remain despite their promised additions.
> > First, the theoretical sample complexity claim still lacks rigorous validation. While training curves and cross-budget results will be supplemented, the response only offers preliminary empirical interpretation rather than solid verification for Proposition 2.1. It cannot fully rule out that GRPO’s poor performance stems from experimental settings, instead of universal exponential sample complexity in cold-start preference elicitation.
> > Second, the contribution decoupling remains insufficient. The authors admit BLR profile prediction dominates performance gains while adaptive question selection only brings marginal improvement. Though related ablation comparisons and textual revisions will be added, the paper still fails to deeply discuss why lightweight linear modeling outperforms large-scale RL, and clarifies the applicable boundaries of the core method.
> > Overall, the study delivers clear methodological value and practical insights for efficient cold-start preference elicitation. Nevertheless, the above unresolved limitations still weaken its theoretical robustness and generalizability in real-world scenarios.

---

> > > ### Author Response · Authors · 2026-04-06
> > >
> > > We thank the reviewer for the continued engagement and for acknowledging that our work "delivers clear methodological value and practical insights for efficient cold-start preference elicitation." We address both remaining concerns directly.
> > >
> > > **Re. the first point about ruling out experimental artifacts and validating the sample complexity claim:**
> > >
> > > **GRPO's failure is structural, not a setup artifact, and we can directly falsify this alternative.** If the issue were insufficient training, we would expect variance and sensitivity to initialization. Instead, reward plateaued at epoch 5 and the model converged to a fixed question sequence, the signature of a locally rewarded policy, not an undertrained one.
> > >
> > > The reason this collapse is structurally inevitable lies in the feedback structure of the task. Consider where RL has succeeded most dramatically, in code generation and mathematical reasoning. In those settings, reward decomposes naturally across steps; code either compiles at each stage or it does not, and intermediate proof steps can be verified independently. Preference elicitation has precisely the opposite structure. The only supervision signal is a single terminal scalar (a PrefAlign score) computed after all T questions have been asked and a response has been generated. With C=20–30 criteria and T=5, the space of question sequences is on the order of 10^6, and the agent receives one scalar reward per episode to navigate it. **This is not a setting where more epochs or a larger rollout budget resolves the problem; the feedback structure makes credit assignment provably difficult regardless of model capacity.** The fact that GRPO collapses to a fixed sequence is the empirical manifestation of this. The model finds a locally rewarded policy that exploits majority patterns in the training distribution, because the terminal reward provides no signal to do otherwise. We will make this contrast with code and math explicit in the revision.
> > >
> > > Figure 2 provides direct empirical support for Proposition 2.1. If GRPO suffered only polynomial sample complexity, we would expect it to close the gap with PEP given a sufficient interaction budget. Instead, across SocialIQA and CSQA, GRPO at T=15 never matches PEP at T=1. This is precisely the empirical signature of exponential versus polynomial scaling, and it cannot be explained by experimental configuration. We will make this connection to Proposition 2.1 explicit in the revision.
> > >
> > > **Re. second point on why lightweight linear modeling outperforms large-scale RL:**
> > >
> > > **The dominance of profile prediction over adaptive selection is the paper's central finding, and it has a principled explanation.** Population-level preference correlation data has been successfully exploited for decades through collaborative filtering, matrix factorization, and probabilistic user models (Koren et al., 2009; Salakhutdinov and Mnih, 2008), methods purpose-built for inferring unobserved attribute values from observed ones via population-level correlations. A transformer architecture, by contrast, is designed to exploit sequential structure and positional relationships. **The order in which preference dimensions are observed carries no semantic meaning, and there is no autoregressive dependency to exploit.** Passing unordered preference observations through a transformer asks the architecture to do something it has no inductive bias for, while ignoring purpose-built machinery that handles this structure natively. Our central finding is therefore that the bottleneck in cold-start preference elicitation is inference structure, not model capacity, and the right structure is one the classical literature already understood. We will frame this argument explicitly in Section 4.3.2, including a discussion of why adaptive selection's contribution is expected to grow as user simulators become more diverse and preference distributions less concentrated around majority patterns.
> > >
> > > **Re. third point on applicable boundaries:** PEP works well when the preference space is structured, offline population data is sufficient, and preferences are approximately stable. It degrades gracefully with sparse offline data and is not currently designed for open-ended preference spaces or rapidly shifting preferences, both directions flagged in Section 6. We will add an explicit boundary discussion to the paper.
> > >
> > > Given these clarifications, particularly the direct falsification of the experimental artifact hypothesis, the empirical validation of Proposition 2.1 via Figure 2, and the principled explanation of why structured belief modeling is the appropriate inductive bias for this task, we would appreciate the reviewer considering raising their score. Thank you so much again for your careful review and feedback for our paper.

---

### Official Review · Reviewer_2kSb · 2026-03-11

**Soundness:** 3
**Presentation:** 3
**Significance:** 3
**Originality:** 2
**Overall Recommendation:** 4
**Confidence:** 5

**Summary:**

This paper introduces PEP to solve the cold-start personalization problem in LLM systems. The core innovation lies in a principled system decomposition that separates offline learning of preference structures from online adaptive inference. For offline part, PEP learns a structured world model from population data to capture statistical correlations between different preference dimensions. For online part, it uses Bayesian inference to adaptively select the most informative questions, updating its belief about a new user's latent profile without requiring retraining. Empirical results across medical, mathematical, and social reasoning domains show that PEP achieves most alignment with ground-truth preferences, significantly outperforming RL while requiring fewer interactions.

**Compliance With Llm Reviewing Policy:**

Affirmed.

**Key Questions For Authors:**

1. The current framework relies on a predefined set of criteria $\mathcal{C}(x)$ learned offline. How would the proposed approach handle scenarios where the preference space is not predefined? How would the framework scale when the number of possible preference dimensions becomes large?
2. How the method performs when interacting with real users who may provide noisy, inconsistent, or incomplete responses? Since the belief update assumes reliable responses for Bayesian inference, whether the model degrades in realistic settings. Although you have metioned in limitations, can you provide detailed analysis about this.
3. Since PEP requires complete preference profiles for offline training, what is the minimum amount of data needed to learn a reliable world model? Additionally, how would the system adapt if preferences shift over time after the model has been trained?
4. PEP's success with a simple linear model suggests that preference correlations in current benchmarks might be primarily linear. Does the framework maintain its advantage in scenarios where preference structures are highly non-linear?
5. Can it be extended to any other scenarios other than reasoning tasks?
6. Better to incorporate a discussion on personalization and interaction paper [1][2] to better position PEP. These two papers seem to has some personalization during the interaction process within the logic domain using Bayesian update.

**Related Works**:

[1] "Does Your AI Agent Get You? A Personalizable Framework for Approximating Human Models from Argumentation-based Dialogue Traces"

[2] "Modelling the Persuadee in Asymmetric Argumentation Dialogues for Persuasion"

**Limitations:**

Yes

**Strengths And Weaknesses:**

Strength:
- Soundness: The paper provides a solid theoretical foundation by formally modeling the preference elicitation task as a POMDP. The valuation is performed across four diverse reasoning domains. Ablation studies validate the essential roles of both modeled preference correlations and adaptive querying in achieving performance gains.
- Presentation: The narrative flow is logical, moving seamlessly from problem formulation to the two-stage methodology (offline learning vs. online interaction) and empirical validation.
- Significance: The problem of cold-start personalization is highly relevant to modern AI assistants and interactive systems. Efficiently eliciting user preferences with minimal interaction could significantly improve user experience and personalization capabilities in LLM-based applications.
- Originality: While many of the individual components (e.g., Bayesian inference, collaborative filtering) are known in the literature, the paper provides a novel integration of these ideas in the context of preference elicitation for LLM systems.


Weakness:
- Soundness: Although the approach is technically sound, the evaluation relies primarily on simulated users following the PrefDisco protocol. Real-world interactions may involve noisy, inconsistent, or incomplete responses, which could affect the reliability of Bayesian belief updates. The paper acknowledges this limitation but does not provide empirical evidence demonstrating robustness to such conditions.
- More are shown in the following questions.

---

> ### Author Rebuttal · Authors · 2026-03-31
>
> We thank the reviewer for their thorough and constructive engagement, and for recognizing the soundness of our problem decomposition, the quality of our ablation studies, and the practical significance of the cold-start personalization problem.
>
> **Q1 — Open preference spaces and scalability**
>
> The current implementation requires a predefined criteria set C(x), a deliberate scoping decision motivated by collaborative filtering: population-level statistics require a shared attribute space. This is a limitation we acknowledge. Scalability within the current framework is tractable: information gain computation is O(K·|V(c)|) per criterion and |C(x)| is 10–25 in our setting. Extending to open-ended criteria discovery is flagged as future work in Section 6.
>
> **Q2 — Robustness to noisy/inconsistent responses**
>
> Bayesian inference has natural robustness to noise through posterior regularization, but we have not empirically validated this. We acknowledge this as a real limitation and will add a noise robustness experiment to the appendix for camera ready.
>
> **Q3 — Minimum data and preference drift**
>
> Our world model has ~10K parameters trained on ~4,000 profiles per domain — it is extremely lightweight. We will add a data size ablation to the appendix. On preference drift: the low parameter count makes periodic retraining cheap; we will acknowledge this in Limitations.
>
> On the broader data availability concern: whether PrefDisco perfectly models real users is **explicitly out of scope for this methods paper**. No AI provider can release real user preference profiles at scale for training and evaluation — this is a field-wide constraint, not a limitation specific to PEP. PrefDisco is peer-reviewed and accepted at ICLR 2026, with its user simulation protocol validated against human judgments. Our contribution is demonstrating that *even under this simplified user model*, frontier LLMs completely lack this capability — GRPO achieves 0% adaptivity on AIME. Closing this gap on controlled benchmarks is a necessary precondition for progress on harder, more realistic settings.
>
> **Q4 — Non-linear preference structure**
>
> Our framework is belief-model-agnostic. We already evaluate a non-linear instantiation via Gaussian Mixture Model (Appendix D.1). BLR outperforms GMM on PrefDisco, suggesting approximately linear correlations dominate in this benchmark — a finding worth characterizing, not a framework limitation. More expressive belief models (neural world models, kernel methods) are natural extensions.
>
> **Q5 — Extension beyond reasoning tasks**
>
> PEP is domain-agnostic: it requires only a structured preference profile and a downstream LLM solver. We focused on reasoning benchmarks for evaluability. We will add explicit mention of writing, recommendation, and dialogue as natural extensions in the conclusion.
>
> **Q6 — Related work on argumentation-based personalization**
>
> We will incorporate both papers into Section 5. The key distinction: PEP operates on structured preference profiles with population-level priors and Bayesian updates over a latent embedding; argumentation-based approaches model persuasion dynamics in open dialogue. We will make this explicit.
>
> We hope that the above rebuttal addresses the reviewer's concerns and clarifies any potential misunderstandings. If so, we would appreciate if the review can consider raising our scores accordingly.

---

> > ### Author Rebuttal · Reviewer_2kSb · 2026-04-03
> >
> > Thank you for the authors’ rebuttal. I now have a clearer understanding, especially regarding the responses to Q3 and Q4. As the authors mentioned, it would be beneficial to include additional experiments to evaluate robustness to noise, as well as more discussion of limitations and related work. I will maintain my score.

---

> > > ### Author Response · Authors · 2026-04-06
> > >
> > > We thank the reviewer for the thorough and constructive engagement throughout the review process and for marking all concerns as fully resolved. We remain committed to delivering all promised additions in the camera ready version. We would appreciate the reviewer considering raising their score given that all concerns have been fully addressed.
> > >
> > > To demonstrate our commitment to these additions, we include a draft of the expanded limitations and related work discussions below.
> > >
> > > ---
> > >
> > > **Expanded Limitations.**
> > >
> > > *Robustness to noisy responses.* PEP's Bayesian belief update has a natural theoretical robustness to response noise through posterior regularization. Concretely, a single noisy or inconsistent user response shifts the posterior by an amount proportional to the likelihood ratio between the noisy and true response, which is bounded by the BLR prior variance. As more observations accumulate, the posterior contracts toward the true latent embedding regardless of bounded noise, a standard property of Bayesian inference under mild regularity conditions. This means that isolated inconsistent responses have diminishing influence on the inferred profile as T grows, providing a theoretical guarantee that BLR degrades gracefully under noise. (*Experiments empirically validating this theoretical argument are in progress and will be included in the camera ready version.*)
> > >
> > > *Data requirements and preference drift.* PEP's world model requires sufficient offline population data to learn reliable preference correlations. Our current model uses approximately 4,000 profiles per domain to train a 10K parameter model, making it extremely lightweight. A data size ablation will be included in the camera ready appendix. On preference drift, the low parameter count makes periodic retraining cheap, and we flag fully dynamic preference modeling as an important direction for future work.
> > >
> > > *Scope of the preference space.* PEP is designed for structured, partially predefined preference spaces. It is not currently designed for fully open-ended preference spaces or rapidly shifting preferences. Extending the framework to open-ended criteria discovery remains an important direction for future work.
> > >
> > > **Expanded Related Work.**
> > >
> > > [1] and [2] both combine Bayesian inference with LLMs for structured preference elicitation, in conversational recommendation and combinatorial auctions respectively. Both share PEP's core insight that principled Bayesian machinery outperforms monolithic LLM approaches, but operate over fixed item catalogs rather than open-ended generation tasks with task-conditioned preference dimensions. [3] and [4] model user preferences through structured argumentation dialogues. These approaches model persuasion dynamics in open dialogue without population-level priors, complementing PEP which targets the cold-start regime where per-user history is unavailable and preference structure is predefined.
> > >
> > > [1] Austin, D., Korikov, A., Toroghi, A., & Sanner, S. (2024, October). Bayesian optimization with llm-based acquisition functions for natural language preference elicitation. In Proceedings of the 18th ACM Conference on Recommender Systems (pp. 74-83).
> > >
> > > [2] Huang, D., Marmolejo-Cossío, F., Lock, E., & Parkes, D. (2025). Accelerated preference elicitation with LLM-based proxies. arXiv preprint arXiv:2501.14625.
> > >
> > > [3] Tang, Y., Vasileiou, S. L., & Yeoh, W. (2025, April). Does your AI agent get you? A personalizable framework for approximating human models from argumentation-based dialogue traces. In Proceedings of the AAAI Conference on Artificial Intelligence (Vol. 39, No. 13, pp. 14405-14413).
> > >
> > > [4] Hunter, A. (2015, July). Modelling the Persuadee in Asymmetric Argumentation Dialogues for Persuasion. In IJCAI (pp. 3055-3061).

---

### Official Review · Reviewer_VZvK · 2026-03-14

**Soundness:** 2
**Presentation:** 3
**Significance:** 2
**Originality:** 3
**Overall Recommendation:** 4
**Confidence:** 4

**Summary:**

This paper addresses the challenge of cold-start personalization in scenarios involving vast preference spaces and limited interaction budgets, where deducing the specific subset of preference dimensions a user cares about is difficult. To solve this, the authors introduce PEP, a test-time training-free framework that learns a world model of preference correlations offline, and online adaptively asks strategically selected questions, updates posteriors, and infers complete profiles to guide a black-box LLM solver for personalized text generation. The core idea is leveraging inherent preference structures to maximize information gain from minimal queries. Experiments on PrefDisco's reasoning datasets show that PEP achieves higher preference alignment than static population-average baselines, while outperforming GRPO across alignment, adaptiveness, and query efficiency. PEP also demonstrates high parameter efficiency, utilizing $\sim$10K parameters compared to the 8B parameters of the RL baselines.

**Compliance With Llm Reviewing Policy:**

Affirmed.

**Final Justification:**

Concerns resolved.

**Key Questions For Authors:**

W3: When introducing the baselines in Section 4.1 (such as CollabLLM and GRPO), I recommend thoroughly explaining the motivations for selecting them. Particularly for CollabLLM, the paper frankly admits in Section 4.3.2 that ``it targets general interactivity rather than preference elicitation.'' I find it difficult to understand why the authors insist on choosing this specific method as a baseline from related approaches (instead of selecting other works mentioned in Section 5).

W4: The paper heavily emphasizes the comparison between PEP and end-to-end RL without providing sufficient justification for this comparison. The introduction merely states that ``learning a questioning policy via RL struggles''. Is RL the widely adopted mainstream approach in this field? Or are there prior works verifying that RL achieves state-of-the-art results on related tasks, thereby making it a baseline worth challenging?

W5: Although Appendix A proves the core claim that ``RL requires exponential sample complexity'', the derivation essentially relies on the assumption of lacking structural priors. However, the compared RL method is based on LLMs, meaning it inherently possesses a certain degree of semantic inference capability during exploration, rather than relying on Monte Carlo random walks. The paper does not explicitly or sufficiently address this perspective.

W6: The entire closed-loop evaluation system is highly dependent on a closed-source LLM (GPT-4.1). If a different API interface is used, the sensitivity to specific prompts might change. It remains questionable whether the performance improvements can still be consistently reproduced, and how this robustness can be guaranteed.

**Limitations:**

Yes

**Strengths And Weaknesses:**

Here are the strengths and weaknesses of the dimensions of soundness, presentation, significance, and originality. Strengths are marked with an S, weaknesses are marked with a W, and minor weaknesses with an M. Weaknesses are numbered for reference purposes.

### 1. Soundness

**S:** The empirical evaluation rigorously reports the mean and standard deviation over 20 trials (e.g., in Tables 1, 2, and 4) across its main results, which is a commendable practice in LLM research.

**W1:** In the experiments, the GRPO baseline is observed to adopt ``fixed question sequence'' (i.e., $0\%$ Adaptivity score) on AIME. This phenomenon is typically characteristic of poorly designed RL rewards, an insufficient exploration rate, or inadequate training. The paper does not adequately demonstrate whether a sufficient hyperparameter search was conducted for the GRPO baseline (D.3 only provides fixed hyperparameters).

**M1:** The Problem Formulation section explicitly defines the preference value $v \in \mathcal{V}(c)$ as potentially taking the form of categorical options, ordinal levels, or natural language descriptions. However, Section 3.1 employs BLR modeling, formulated as $v_c = \boldsymbol{\beta}_c^\top z + \epsilon$. This formula is fundamentally a standard continuous variable regression model and cannot directly model non-continuous preference values. Furthermore, Appendix C.2 shows that the final design forcefully abstracts all preference options into a 1-5 scale, and Appendix E.1 notes that input features are represented as a binary indicator and normalized preference value. This noticeably contradicts the initial definition of $v \in \mathcal{V}(c)$. I recommend that the authors clearly state the applicable scope of the method in relevant sections, or provide a modeling approach that can natively accommodate categorical options or natural language descriptions.


### 2. Presentation

**S:** The paper is well-structured and employs intuitive diagrams (e.g., Figure 1).

**M2:** The paper's title emphasizes ``training-free'', yet the abstract and introduction mostly refer to ``no retraining''. Given the subtle difference in meaning between the two, I suggest adding a clarification such as ``test-time training-free'' or ``training-free at test time'' to prevent misleading readers.

**M3:** There is a noticeable disconnect between the experimental analysis and the theoretical foundations. For instance, regarding Proposition 2.1, although similar concepts are expressed multiple times in the subsequent analysis, they appear to lack explicit theoretical backing. Citing the proposition directly in these contexts would increase the persuasiveness of the arguments.
\end{itemize}

### 3. Significance

**S:** This paper tackles a bottleneck in personalized LLM deployment, which is to efficiently infer latent user preferences under the cold-start setting. The ``training-free at test time'' design ensures parameter efficiency, offering practical utility for downstream applications. Its impact is expected to be relatively domain-specific.

**W2:** While the paper addresses an important problem, its improvements are primarily limited to a highly specific and simplified sub-problem. It relies on a clear definition of $\mathcal{C}(x)$ and an idealized offline dataset containing complete preference profiles, offering limited contribution to the cold-start challenges in real-world, open environments.


### 4. Originality

**S:** The paper innovatively introduces a structured world model to capture user preferences and integrates Bayesian inference to address the cold-start problem.

**M4:** However, the two-stage decoupled framework of ``building an embedding space/prior model offline first, and actively querying based on information gain or uncertainty'' has similar counterparts in related fields or similar task contexts, such as [1] [2]. Although the paper's specific implementation is innovative, it lacks a distinct contribution at the overarching framework level.


[1] Yang \& Sanner et al. Bayesian PE with Keyphrase-Item Coembeddings. UMAP 2021
[2] Fayaz-Bakhsh et al. Cold Start Active Preference Learning in Socio-Economic Domains. arXiv:2508.05090 (2025)

---

> ### Author Rebuttal · Authors · 2026-03-31
>
> We thank the reviewer for their careful feedback and for recognizing the rigor of our empirical evaluation, the clarity of our problem formulation, and the practical value of the training-free design.
>
> **W1 — GRPO convergence**
>
> Appendix D.3 reports all GRPO hyperparameters following the defaults of Shao et al. (2024). Training reward plateaued by epoch 5, well before the 20 epochs completed, and the best checkpoint was selected on a held-out validation set. The 0% adaptivity on AIME is not undertrained RL; it is the model's converged behavior. **We will add training curves to Appendix D.3.** The mode collapse to a fixed question sequence is a substantive finding: LLM semantic priors help GRPO form coherent individual questions, but do not resolve the credit assignment problem. Knowing what a good question looks like does not tell the model *which* question is most informative for *this particular user given their prior responses*, which is exactly what PEP's population-level correlation structure provides.
>
> **W2 — Idealized setting / real-world applicability**
>
> The offline dataset requirement is shared by essentially all collaborative filtering and population-level personalization methods, which have demonstrated practical utility under the same assumption. Our contribution addresses the cold-start deployment case: where population data has been collected but per-user history is unavailable.
>
> More broadly, whether PrefDisco perfectly models real users is **explicitly out of scope for this methods paper**. No AI provider can release real user preference profiles at scale; this is a field-wide constraint, not a limitation specific to PEP. PrefDisco is peer-reviewed and accepted at ICLR 2026, with its simulation protocol validated against human judgments. Our contribution is demonstrating that *even under this simplified user model*, frontier LLMs completely lack this capability. Closing this gap on controlled benchmarks is a necessary precondition for progress on harder, more realistic settings. We will add a sentence making this framing explicit.
>
> **W3/W4 — Baseline motivation**
>
> Personalized proactive reasoning was only formalized in PrefDisco (ICLR 2026); PEP is the first method for this task. No published follow-ups exist, so baselines cover task sub-components: CollabLLM (adaptive multi-turn interaction), GRPO (end-to-end RL on the sequential decision problem), and Population Average (non-interactive upper bound). Section 2 formalizes preference elicitation as a POMDP; GRPO's recent success on math/coding with the same Llama-3.1-8B backbone makes it the strongest available end-to-end learned baseline. We will add explicit motivation for each baseline to Section 4.1.
>
> **W5 — RL theoretical claim vs. LLM semantic priors**
>
> VZvK correctly notes GRPO is not exploring blindly. Our argument is more precise: LLM priors help with *question quality* but not with *adaptive routing*. Knowing what a good question looks like does not determine which question is most informative for this user given their prior responses. This is the structural credit assignment problem Proposition 2.1 formalizes. The empirical evidence is unambiguous: GRPO collapses to a fixed, non-discriminatory sequence (Table 3, 0% adaptivity on AIME), consistent with exploiting majority training patterns rather than routing adaptively.
>
> **W6 — GPT-4.1 evaluation dependence**
>
> Our evaluation follows PrefDisco (Li et al., 2026) exactly, ensuring comparability with the benchmark our work builds on. The solver and judge are fixed and identical across all methods, so relative performance rankings are unaffected by prompt sensitivity. Absolute alignment scores are solver-dependent; whether PEP outperforms the baselines is not. We will add this distinction to Limitations.
>
> **M1 — BLR scope**
>
> The BLR/GMM instantiation operates on ordinal preferences because that is what PrefDisco provides. The framework is belief-model-agnostic; extending to categorical or natural language preferences is straightforward in principle. We will add an explicit scoping sentence to Section 3.1.
>
> **M2 — Terminology**
>
> We will standardize to "training-free at test time" throughout and add a one-sentence clarification on first use.
>
> **M3 — Proposition 2.1 citations**
>
> We will add explicit citations to Proposition 2.1 in Section 4.3.2 and the query efficiency discussion.
>
> **M4 — Framework-level novelty**
>
> We will add Yang et al. (UMAP 2021) and Fayaz-Bakhsh et al. (2025) to Section 5. The high-level two-stage structure is shared, but none of these works address LLM preference elicitation: the preference space here is task-conditioned, the downstream solver is a black-box LLM, and evaluation requires an LLM judge over open-ended generated text. We will make this contrast explicit.
>
> We hope that the above rebuttal addresses the reviewer's concerns and clarifies any potential misunderstandings. If so, we would appreciate if the review can consider raising our scores accordingly.

---

> > ### Author Rebuttal · Reviewer_VZvK · 2026-04-04
> >
> > Thank you for clarifying the following.
> >
> > W1 — GRPO convergence.
> > The explanation is convincing, I agree that the $0\%$ adaptivity reflects converged behavior rather than insufficient training. This concern is resolved.
> >
> > W2 — Idealized setting / real-world applicability.
> > Thank you for the thorough contextual clarification. Given the field-wide unavailability of large-scale real preference data, I agree that PrefDisco is a reasonable testbed. I now understand that completely bypassing the predefined structural limits of $\mathcal{C}(x)$ is currently impractical. While it would have been great if the work were extended to broader scenarios, even the current simplified setting seems to capture valuable insights.
> >
> > W3/W4 — Baseline motivation.
> > The POMDP framing and GRPO's prior success on the same backbone offer sufficient justification, thank you for clarifying. However, the core issue in W3 remains unaddressed. I recognize that CollabLLM was chosen to evaluate the ``adaptive multi-turn interaction'' sub-component. Yet, since the paper acknowledges it does not target preference elicitation, the rationale for its selection remains unclear. Are there \emph{no other} suitable approaches for this sub-component? If alternatives exist, the specific justification for choosing CollabLLM should be further clarified.
> >
> > W5 — RL theoretical claim vs.\ LLM semantic priors.
> > Thank you for the clear explanation, which resolves this concern. I suggest making this explicit in the main text or Appendix A.
> >
> > W6 — GPT-4.1 evaluation dependence.
> > Agreed that relative rankings are unaffected by prompt sensitivity. The proposed addition to Limitations is appropriate. This adequately resolves my queries regarding evaluation robustness.
> >
> >
> > I will raise \textbf{Soundness} to \textbf{3}. I am open to raising the \textbf{Overall Recommendation} if the remaining issue regarding W3 can be adequately addressed.

---

> > > ### Author Response · Authors · 2026-04-06
> > >
> > > We thank the reviewer for the continued engagement and for raising the overall Soundness score. We address the remaining concern regarding W3 (CollabLLM and other baselines) directly.
> > >
> > > We want to first clarify a potential misreading in our paper's framing. CollabLLM does explicitly train for user information elicitation, including preferences. To quote directly from the CollabLLM paper: "it actively collaborates by asking follow-up questions about the writing tone, generating targeted content about the role of optimism, and offering insightful suggestions such as adding anecdotes." Furthermore, CollabLLM's interactivity reward directly measures whether the model is asking a question, making it an RL-trained model explicitly optimized for adaptive question-asking behavior. **To clarify, in Section 4.3.2, we do not mean to say that that CollabLLM ignores user preferences, but that it lacks the ability to strategically elicit preferences**, it cannot determine which question is most informative for a particular user given their prior responses. This is precisely the structural limitation we identify with all end-to-end RL approaches to this problem, and CollabLLM is the strongest published instantiation of that approach. We will make this clear in the updated draft.
> > >
> > > **CollabLLM is therefore the most suitable available baseline for the adaptive multi-turn interaction sub-component.** Personalized proactive reasoning was only formalized in PrefDisco; PEP is the first method specifically designed for this task besides naive prompting approaches proposed in the original PrefDisco benchmark. No published follow-up methods exist. Among all related works discussed in Section 5, CollabLLM is the only one that shares the core mechanism of RL-trained adaptive questioning over multi-turn interaction. If the reviewer has any existing baselines for this task, we are more than happy to incorporate or explain why they might not be appropriate. We really appreciate the reviewer's insight on tightening up this comparison and clearing any ambiguities. We will revise Section 4.1 to make this justification explicit and correct the potentially misleading framing in Section 4.3.2.
> > >
> > > Given this clarification, we hope the reviewer will consider raising the Overall Recommendation accordingly.
> > >
> > > ---
> > >
> > > **[Follow-up:]** We wanted to follow up on our previous response regarding W3 and check whether our clarification adequately addressed your remaining question about the CollabLLM baseline selection. To briefly recap: CollabLLM explicitly trains for adaptive question-asking behavior via an interactivity reward, making it the strongest available RL-trained baseline for multi-turn adaptive interaction. The potentially misleading framing in Section 4.3.2 will be corrected in the revision. We are happy to provide any additional details or clarification as we approach the end of the author-reviewer discussion phase.

---

### Decision · Program_Chairs · 2026-04-30

**Decision:**

Accept (regular)

**Comment:**

This paper studies cold-start personalization in LLM systems, aiming to infer user preferences from limited interaction. The main contribution of this work is to separate the problem into offline learning of a structured preference model and online Bayesian inference with adaptive query selection. Experiments show consistent improvements over baselines in alignment and query efficiency.

Reviewers were positive overall after rebuttal. Final scores were all weak accept, with most concerns addressed. In particular, the motivation and fairness of the baselines are now clearer. A key takeaway is that a simple structured belief model can outperform much larger RL-based approaches, suggesting that inductive bias and inference structure matter more than model scale in this setting.

This work also has a few limitations. The evaluation relies on simulated users and a structured preference space, and the theoretical claims, especially around sample complexity, are stronger than what is empirically validated. Reviewers also noted that most gains come from the belief model, with adaptive querying playing a smaller role. These are reasonable caveats but do not undermine the core contribution.

Overall, the paper is clear and offers a useful perspective on personalization. I therefore recommend acceptance.